# NAISR: A 3D Neural Additive Model for Interpretable Shape Representation

**Yining Jiao**[1], **Carlton Zdanski**[1], **Julia Kimbell**[1], **Andrew Prince**[1], **Cameron Worden**[1],
**Samuel Kirse**[2], **Christopher Rutter**[3], **Benjamin Shields**[1], **William Dunn**[1], **Jisan Mahmud**[1],
**Marc Niethammer**[1]*; **for the Alzheimer's Disease Neuroimaging Initiative**[†]
[1]University of North Carolina at Chapel Hill, [2]Wake Forest School of Medicine,
[3]The Ohio State University College of Medicine

## Abstract

Deep implicit functions (DIFs) have emerged as a powerful paradigm for many computer vision tasks such as 3D shape reconstruction, generation, registration, completion, editing, and understanding. However, given a set of 3D shapes with associated covariates there is at present no shape representation method which allows to precisely represent the shapes while capturing the individual dependencies on each covariate. Such a method would be of high utility to researchers to discover knowledge hidden in a population of shapes. For scientific shape discovery, we propose a 3D Neural Additive Model for Interpretable Shape Representation (NAISR) which describes individual shapes by deforming a shape atlas in accordance to the effect of disentangled covariates. Our approach captures shape population trends and allows for patient-specific predictions through shape transfer. NAISR is the first approach to combine the benefits of deep implicit shape representations with an atlas deforming according to specified covariates. We evaluate NAISR with respect to shape reconstruction, shape disentanglement, shape evolution, and shape transfer on three datasets: 1) *Starman*, a simulated 2D shape dataset; 2) the ADNI hippocampus 3D shape dataset; and 3) a pediatric airway 3D shape dataset. Our experiments demonstrate that NAISR achieves excellent shape reconstruction performance while retaining interpretability. Our code is publicly available.

## 1 Introduction

Deep implicit functions (DIFs) have emerged as efficient representations of 3D shapes (Park et al., 2019; Novello et al., 2022; Mescheder et al., 2019; Yang et al., 2021), deformation fields (Wolterink et al., 2022b), images and videos (Sitzmann et al., 2020), graphs, and manifolds (Grattarola & Vandergheynst, 2022). For example, DeepSDF (Park et al., 2019) represents shapes as the level set of a signed distance field (SDF) with a neural network. In this way, 3D shapes are compactly represented as continuous and differentiable functions with infinite resolution. In addition to representations of geometry such as voxel grids (Wu et al., 2016; 2015; 2018), point clouds (Achlioptas et al., 2018; Yang et al., 2018; Zamorski et al., 2020) and meshes (Groueix et al., 2018; Wen et al., 2019; Zhu et al., 2019), DIFs have emerged as a powerful paradigm for many computer vision tasks. DIFs are used for 3D shape reconstruction (Park et al., 2019; Mescheder et al., 2019; Sitzmann et al., 2020), generation (Gao et al., 2022), registration (Deng et al., 2021; Zheng et al., 2021; Sun et al., 2022a; Wolterink et al., 2022b), completion (Park et al., 2019), editing (Yang et al., 2022a) and understanding (Palafox et al., 2022).

Limited attention has been paid to shape analysis with DIFs. Specifically, given a set of 3D shapes with a set of covariates attributed to each shape, a shape representation method is still desired which

---

*Corresponding author.

†Data used in preparation of this article were obtained from the Alzheimer's Disease Neuroimaging Initiative (ADNI) database (adni.loni.usc.edu). As such, the investigators within the ADNI contributed to the design and implementation of ADNI and/or provided data but did not participate in analysis or writing of this report. A complete listing of ADNI investigators can be found at: http://adni.loni.usc.edu/wp-content/uploads/how_to_apply/ADNI_Acknowledgement_List.pdf

can precisely represent shapes and capture dependencies among a set of shapes. There is currently no shape representation method that can quantitatively capture how covariates geometrically and temporally affect a population of 3D shapes; neither on average nor for an individual. However, capturing such effects is desirable as it is often difficult and sometimes impossible to control covariates (such as age, sex, and weight) when collecting data. Further, understanding the effect of such covariates is frequently a goal of medical studies. Therefore, it is critical to be able to disentangle covariate shape effects on the individual and the population-level to better understand and describe shape populations. Our approach is grounded in the estimation of a shape atlas (i.e., a template shape) whose deformation allows to capture covariate effects and to model shape differences. Taking the airway as an example, a desired atlas representation should be able to answer the following questions:

- Given an atlas shape, how can one accurately represent shapes and their dependencies?
- Given the shape of an airway, how can one disentangle covariate effects from each other?
- Given a covariate, e.g., age, how does an airway atlas change based on this covariate?
- Given a random shape, how will the airway develop after a period of time?

To answer these questions, we propose a Neural Additive Interpretable Shape Representation (NAISR), an interpretable way of modeling shapes associated with covariates via a shape atlas. Table 1 compares NAISR to existing shape representations with respect to the following properties:

- **Implicit** relates to how a shape is described. Implicit representations are desirable as they naturally adapt to different resolutions of a shape and also allow shape completion (i.e., reconstructing a complete shape from a partial shape, which is common in medical scenarios) with no additional effort.

- **Deformable** captures if a shape representation results in point correspondences between shapes, e.g., via a displacement field. Specifically, we care about point correspondences between the target shapes and the atlas shape. A deformable shape representation helps to relate different shapes.

- **Disentangleable** indicates whether a shape representation can disentangle individual covariate effects for a shape. These covariate-specific effects can then be composed to obtain the overall displacement of an atlas/template shape.

- **Evolvable** denotes whether a shape representation can evolve the shape based on changes of a covariate, capturing the influence of *individual* covariates on the shape. An evolvable atlas statistically captures how each covariate influences the shape population, e.g., in scientific discovery scenarios.

- **Transferable** indicates whether shape changes can be transferred to a given shape. E.g., one might want to edit an airway based on a simulated surgery and predict how such a surgical change manifests later in life.

- **Interpretable** indicates a shape representation that is simultaneously *deformable*, *disentangleable*, *evolvable*, and *transferable*. Such an interpretable model is applicable to tasks ranging from the estimation of disease progression to assessing the effects of normal aging or weight gain on shape.

NAISR is the first implicit shape representation method to investigate an atlas-based representation of 3D shapes in a deformable, disentangleable, transferable and evolvable way. To demonstrate the generalizability of NAISR, we test NAISR on a simulated dataset and two realistic medical datasets [*]: 1) *Starman*, a simulated 2D shape dataset (Bône et al., 2020); 2) the ADNI hippocampus 3D shape dataset (Jack Jr et al., 2008); and 3) a pediatric airway 3D shape dataset. We quantitatively demonstrate superior performance over the baselines.

## 2 RELATED WORK

We introduce the three most related research directions here. A more comprehensive discussion of related work is available in Section S.1 of the supplementary material.

---

[*]Medical shape datasets are our primary choice because quantitative shape analysis is a common need for scientific discovery for such datasets.

| Method | Implicit | Deformable | Disentangleable | Evolvable | Transferable | Interpretable |
|---|---|---|---|---|---|---|
| ConditionalTemplate (Dalca et al., 2019) | ✗ | ✓ | ✗ | ✓ | ✗ | ✗ |
| 3DAttriFlow (Wen et al., 2022) | ✗ | ✓ | ✗ | ✓ | ✗ | ✗ |
| DeepSDF (Park et al., 2019) | ✓ | ✗ | ✗ | ✗ | ✗ | ✗ |
| A-SDF (Mu et al., 2021) | ✓ | ✗ | ✗ | ✓ | ✓ | ✗ |
| DIT (Zheng et al., 2021), DIF (Deng et al., 2021), NDF (Sun et al., 2022a) | ✓ | ✓ | ✗ | ✗ | ✗ | ✗ |
| NASAM (Wei et al., 2022) | ✓ | ✓ | ✗ | ✓ | ✗ | ✗ |
| Ours (NAISR) | ✓ | ✓ | ✓ | ✓ | ✓ | ✓ |

Table 1: Comparison of shape representations based on the desirable properties discussed in Section 1. A ✓ indicates that a model has a property; a ✗ indicates that it does not. Only NAISR has all the desired properties.

**Deep Implicit Functions.**   Compared with geometry representations such as voxel grids (Wu et al., 2016; 2015; 2018), point clouds (Achlioptas et al., 2018; Yang et al., 2018; Zamorski et al., 2020) and meshes (Groueix et al., 2018; Wen et al., 2019; Zhu et al., 2019), DIFs are able to capture highly detailed and complex 3D shapes using a relatively small amount of data (Park et al., 2019; Mu et al., 2021; Zheng et al., 2021; Sun et al., 2022a; Deng et al., 2021). They are based on classical ideas of level set representations (Sethian, 1999; Osher & Fedkiw, 2005); however, whereas these classical level set methods represent the level set function on a grid, DIFs parameterize it as a *continuous function*, e.g., by a neural network. Hence, DIFs are not reliant on meshes, grids, or a discrete set of points. This allows them to efficiently represent natural-looking surfaces (Gropp et al., 2020; Sitzmann et al., 2020; Niemeyer et al., 2019). Further, DIFs can be trained on a diverse range of data (e.g., shapes and images), making them more versatile than other shape representation methods. This makes them useful in applications ranging from computer graphics, to virtual reality, and robotics (Gao et al., 2022; Yang et al., 2022a; Phongthawee et al., 2022; Shen et al., 2021). *We therefore formulate NAISR based on DIFs.*

**Neural Deformable Models**   Neural Deformable Models (NDMs) establish point correspondences with DIFs. In computer graphics, there has been increasing interest in NDMs to animate scenes (Liu et al., 2022; Bao et al., 2023; Zheng et al., 2023), objects (Lei & Daniilidis, 2022; Bao et al., 2023; Zhang et al., 2023), and digital humans (Park et al., 2021b; Zhang & Chen, 2022; Niemeyer et al., 2019). Establishing point correspondences is also important to help experts to detect, understand, diagnose, and track diseases. NDMs have shown to be effective in exploring point correspondences for medical images (Han et al., 2023b; Tian et al., 2023; Wolterink et al., 2022a; Zou et al., 2023) and shapes (Sun et al., 2022a; Yang et al., 2022b) Among the NDMs for shape representations, ImplicitAtlas (Yang et al., 2022b), DIF-Net (Deng et al., 2021), DIT (Zheng et al., 2021), and NDF (Sun et al., 2022a) capture point correspondences between target and template shapes within NDMs. Currently, no continuous deformable shape representation which models the effects of covariates exists. *NAISR provides a model with such capabilities.*

**Explainable Artificial Intelligence.**   The goal of eXplainable Artificial Intelligence (XAI) is to provide human-understandable explanations for decisions and actions of an AI model. Various flavors of XAI exist, including counterfactual inference (Berrevoets et al., 2021; Moraffah et al., 2020; Thiagarajan et al., 2020; Chen et al., 2022), attention maps (Zhou et al., 2016; Jung & Oh, 2021; Woo et al., 2018), feature importance (Arik & Pfister, 2021; Ribeiro et al., 2016; Agarwal et al., 2020), and instance retrieval (Crabbe et al., 2021). NAISR is inspired by neural additive models (NAMs) (Agarwal et al., 2020) which in turn are inspired by generalized additive models (GAMs) (Hastie, 2017). NAMs are based on a linear combination of neural networks each attending to a *single* input feature, thereby allowing for interpretability. NAISR extends this concept to interpretable 3D shape representations. This is significantly more involved as, unlike for NAMs and GAMs, we are no longer dealing with scalar values, but with 3D shapes. *NAISR will provide interpretable results by capturing spatial deformations with respect to an estimated atlas shape such that individial covariate effects can be distinguished.*

## 3   METHOD

This section disccues our NAISR model and how we obtain the desired model properties of Section 1.

### 3.1   PROBLEM DESCRIPTION

Consider a set of shapes $\mathcal{S} = \{\mathcal{S}^k\}$ where each shape $\mathcal{S}^k$ has an associated vector of covariates $\mathbf{c} = [c_1, ..., c_i, ..., c_N]$ (e.g., age, weight, sex). Suppose $\{\mathcal{S}^k\}$ are well pre-aligned and centered (e.g., based on Iterative Closest Point (ICP) registration (Arun et al., 1987) or landmark registration; see

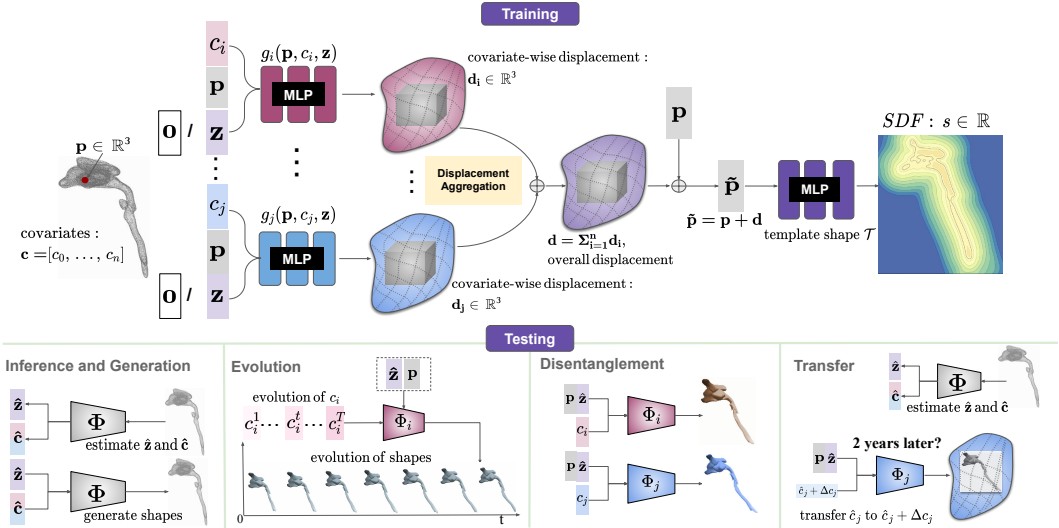

Figure 1: **Neural Additive Implicit Shape Representation.** During training we learn the template $\mathcal{T}$ and the multi-layer perceptrons (MLPs) $\{g_i\}$ predicting the covariate-wise displacement fields $\{\mathbf{d}_i\}$. The displacement fields are added to obtain the overall displacement field $\mathbf{d}$ defined in the target space; $\mathbf{d}$ provides the displacement between the deformed template shape $\mathcal{T}$ and the target shape. Specifically the template shape is queried not at its original coordinates $\mathbf{p}$, but at $\tilde{\mathbf{p}} = \mathbf{p} + \mathbf{d}$ effectively spatially deforming the template. At test time we evaluate the trained MLPs for shape reconstruction, evolution, disentanglement, and shape transfer.

Section S.2 for details). Our goal is to obtain a representation which describes the entire set $\mathcal{S}$ while accounting for the covariates. Our approach estimates a template shape, $\mathcal{T}$ (the shape atlas), which approximates $\mathcal{S}$. Specifically, $\mathcal{T}$ is deformed based on a set of displacement fields $\{\mathcal{D}^k\}$ such that the individual shapes $\{\mathcal{S}^k\}$ are approximated well by the transformed template.

A displacement field $\mathcal{D}^k$ describes how a shape is related to the template shape $\mathcal{T}$. The factors that cause this displacement may be directly observed or not. For example, observed factors may be covariates such as subject age, weight, or sex; i.e., $\mathbf{c}_k$ for subject $k$. Factors that are not directly observed may be due to individual shape variation, unknown or missing covariates, or variations due to differences in data acquisition or data preprocessing errors. We capture these not directly observed factors by a latent code $\mathbf{z}$. The covariates $\mathbf{c}$ and the latent code $\mathbf{z}$ then contribute jointly to the displacement $\mathcal{D}$ with respect to the template shape $\mathcal{T}$.

Inspired by neural additive (Agarwal et al., 2020) and generalized additive (Hastie, 2017) models, we assume the overall displacement field is the sum of displacement fields that are controlled by individual covariates: $\mathcal{D} = \Sigma_i \mathcal{D}_i$. Here, $\mathcal{D}_i$ is the displacement field controlled by the i-th covariate, $c_i$. This results by construction in an overall displacement $\mathcal{D}$ that is disentangled into several sub-displacement fields $\{\mathcal{D}_i\}$.

## 3.2 Model Formulation

Figure 1 gives an overview of NAISR. To obtain a continuous atlas representation, we use DIFs to represent both the template $\mathcal{T}$ and the displacement field $\mathcal{D}$. The template shape $\mathcal{T}$ is represented by a signed distance function, where the zero level set $\{\mathbf{p} \in \mathbb{R}^3 | \mathcal{T}(\mathbf{p}) = 0\}$ captures the desired template shape. A displacement $\mathcal{D}_i$ of a particular point $\mathbf{p}$ can also be represented as a function $\mathbf{d}_i = f_i(\mathbf{p}, c_i, \mathbf{z}) \in \mathbb{R}^3$. We use SIREN (Sitzmann et al., 2020) as the backbone for $\mathcal{T}(\cdot)$ and $\{f_i(\cdot)\}$. Considering that the not directly observed factors might influence the geometry of all covariate-specific networks, we make the latent code, $\mathbf{z}$, visible to all subnetworks $\{f_i(\cdot)\}$. We normalize the covariates so that they are centered at zero. To assure that a zero covariate value results in a zero displacement we parameterize the displacement fields as $\mathbf{d}_i = g_i(\mathbf{p}, c_i, \mathbf{z})$ where

$$g_i(\mathbf{p}, c_i, \mathbf{z}) = f_i(\mathbf{p}, c_i, \mathbf{z}) - f_i(\mathbf{p}, 0, \mathbf{z}) . \tag{1}$$

The sub-displacement fields are added to obtain the overall displacement field

$$\mathbf{d} = g(\mathbf{p}, \mathbf{c}, \mathbf{z}) = \sum_{i=1}^{N} g_i(\mathbf{p}, c_i, \mathbf{z}) . \tag{2}$$

We then deform the template shape $\mathcal{T}$ to obtain an implicit representation of a target shape

$$s = \Phi(\mathbf{p}, \mathbf{c}, \mathbf{z}) = \mathcal{T}(\tilde{\mathbf{p}}) = \mathcal{T}\left(\mathbf{p} + \sum_{i=1}^{N} g_i(\mathbf{p}, c_i, \mathbf{z})\right), \qquad (3)$$

where $\mathbf{p}$ is a point in the source shape space, e.g., a point on the surface of shape $\mathcal{S}_i$ and $\tilde{\mathbf{p}}$ represents a point in the template shape space, e.g., a point on the surface of the template shape $\mathcal{T}$. To investigate how an individual covariate $c_i$ affects a shape we can simply extract the zero level set of

$$s_i = \Phi_i(\mathbf{p}, c_i, \mathbf{z}) = \mathcal{T}(\mathbf{p} + \mathbf{d}_i) = \mathcal{T}(\mathbf{p} + g_i(\mathbf{p}, c_i, \mathbf{z})). \qquad (4)$$

### 3.3 TRAINING

**Losses.** All our losses are ultimately summed over all shapes of the training population with the appropriate covariates $\mathbf{c}_k$ and shape code $\mathbf{z}_k$ for each shape $\mathcal{S}^k$. For ease of notation, we describe them for individual shapes. For each shape, we sample on-surface and off-surface points. On-surface points have zero signed distance values and normal vectors extracted from the gold standard[†] mesh. Off-surface points have non-zero signed distance values but no normal vectors. Our reconstruction losses follow Sitzmann et al. (2020); Novello et al. (2022). For points on the surface, the losses are

$$\mathcal{L}_{\text{on}}(\Phi, \mathbf{c}, \mathbf{z}) = \int_{\mathcal{S}} \lambda_1 \underbrace{\|\|\nabla_{\mathbf{p}}\Phi(\mathbf{p}, \mathbf{c}, \mathbf{z})| - 1\|}_{\mathcal{L}_{\text{Eikonal}}} + \lambda_2 \underbrace{\|\Phi(\mathbf{p}, \mathbf{c}, \mathbf{z})\|}_{\mathcal{L}_{\text{Dirichlet}}} + \lambda_3 \underbrace{(1 - \langle \nabla_{\mathbf{p}}\Phi(\mathbf{p}, \mathbf{c}, \mathbf{z}), \mathbf{n}(\mathbf{p})\rangle)}_{\mathcal{L}_{\text{Neumann}}} d\mathbf{p}, \qquad (5)$$

where $\mathbf{n}(\mathbf{p})$ is the normal vector at $\mathbf{p}$ and $\langle\cdot\rangle$ denotes cosine similarity. For off-surface points, we use

$$\mathcal{L}_{\text{off}}(\Phi, \mathbf{c}, \mathbf{z}) = \int_{\Omega\setminus\mathcal{S}} \lambda_4 \underbrace{|\Phi(\mathbf{p}, \mathbf{c}, \mathbf{z}) - s_{tgt}(\mathbf{p})|}_{\mathcal{L}_{\text{Dirichlet}}} + \lambda_5 \underbrace{\|\|\nabla_{\mathbf{p}}\Phi(\mathbf{p}, \mathbf{c}, \mathbf{z})| - 1\|}_{\mathcal{L}_{\text{Eikonal}}} d\mathbf{p}, \qquad (6)$$

where $s_{tgt}(\mathbf{p})$ is the signed distance value at $\mathbf{p}$ corresponding to a given target shape. Similar to (Park et al., 2019; Mu et al., 2021) we penalize the squared $L^2$ norm of the latent code $z$ as

$$\mathcal{L}_{\text{lat}}(\mathbf{z}) = \lambda_6 \frac{1}{\sigma^2} \|z\|_2^2. \qquad (7)$$

As a result, our overall loss (for a given shape) is

$$\mathcal{L}(\Phi, \mathbf{c}, \mathbf{z}) = \underbrace{\mathcal{L}_{\text{lat}}(\mathbf{z})}_{\text{as regularizer}} + \underbrace{\mathcal{L}_{\text{on}}(\Phi, \mathbf{c}, \mathbf{z}) + \mathcal{L}_{\text{off}}(\Phi, \mathbf{c}, \mathbf{z})}_{\text{for reconstrution}}, \qquad (8)$$

where the parameters of $\Phi$ and $\mathbf{z}$ are trainable.

### 3.4 TESTING

As shown in Figure 1, our proposed `NAISR` is designed for shape reconstruction, shape disentanglement, shape evolution, and shape transfer.

**Shape Reconstruction and Generation.** As illustrated in the inference section in Figure 1, a shape $s_{tgt}$ is given and the goal is to recover its corresponding latent code $\mathbf{z}$ and the covariates $\mathbf{c}$. To estimate these quantities, the network parameters stay fixed and we optimize over the covariates $\mathbf{c}$ and the latent code $\mathbf{z}$ which are both randomly initialized (Park et al., 2019; Mu et al., 2021). Specifically, we solve the optimization problem

$$\hat{\mathbf{c}}, \hat{\mathbf{z}} = \arg\min_{\mathbf{c}, \mathbf{z}} \mathcal{L}(\Phi, \mathbf{c}, \mathbf{z}). \qquad (9)$$

In clinical scenarios, the covariates $\mathbf{c}$ might be known (e.g., recorded age or weight at imaging time). In this case, we only infer the latent code $\mathbf{z}$ by the optimization

$$\hat{\mathbf{z}} = \arg\min_{\mathbf{z}} \mathcal{L}(\Phi, \mathbf{c}, \mathbf{z}). \qquad (10)$$

A new patient shape with different covariates can be generated by extracting the zero level set of $\Phi(\mathbf{p}, \mathbf{c}_{\text{new}}, \hat{\mathbf{z}})$.

---

[†]In medical imaging, there is typically no groundtruth. We use *gold standard* to indicate shapes based off of manual or automatic segmentations, which are our targets for shape reconstruction.

**Shape Evolution.** Shape evolution along covariates $\{c_i\}$ is desirable in shape analysis to obtain knowledge of disease progression or population trends in the shape population $\mathcal{S}$. For a time-varying covariate $(c_i^0, ..., c_i^t, ..., c_i^T)$, we obtain the corresponding shape evolution by $(\Phi_i(\mathbf{p}, c_i^0, \hat{\mathbf{z}}), ..., \Phi_i(\mathbf{p}, c_i^t, \hat{\mathbf{z}}), ..., \Phi_i(\mathbf{p}, c_i^T, \hat{\mathbf{z}}))$. If some covariates are correlated (e.g., age and weight), we can first obtain a reasonable evolution of the covariates $(\mathbf{c}^0, ..., \mathbf{c}^t, ..., \mathbf{c}^T)$ and the corresponding shape evolution as $(\Phi(\mathbf{p}, \mathbf{c}^0, \hat{\mathbf{z}}), ..., \Phi(\mathbf{p}, \mathbf{c}^t, \hat{\mathbf{z}}), ..., \Phi(\mathbf{p}, \mathbf{c}^T, \hat{\mathbf{z}}))$. By setting $\hat{\mathbf{z}}$ to $\mathbf{0}$, one can observe how a certain covariate influences the shape population on average.

**Shape Disentanglement.** As shown in the disentanglement section in Figure 1, the displacement for a particular covariate $c_i$ displaces point $\mathbf{p}$ in the source space to $\mathbf{p} + \mathbf{d}_i$ in the template space for a given or inferred $\hat{\mathbf{z}}$ and $c_i$. We obtain the corresponding signed distance field as

$$s_i = \Phi_i(\mathbf{p}, c_i, \hat{\mathbf{z}}) = \mathcal{T}(\mathbf{p} + \mathbf{d}_i) = \mathcal{T}(\mathbf{p} + g_i(\mathbf{p}, c_i, \hat{\mathbf{z}})). \tag{11}$$

As a result, the zero level sets of $\{\Phi_i(\cdot)\}$ represent shapes warped by the sub-displacement fields controlled by $c_i$.

**Shape Transfer.** We use the following clinical scenario to introduce the shape transfer task. Suppose a doctor has simulated a surgery on an airway shape with the goal of previewing treatment effects on the shape after a period of time. This question can be answered by our shape transfer approach. Specifically, as shown in the transfer section in Figure 1, after obtaining the inferred latent code $\hat{\mathbf{z}}$ and covariates $\hat{\mathbf{c}}$ from reconstruction, one can transfer the shape from the current covariates $\hat{\mathbf{c}}$ to new covariates $\hat{\mathbf{c}} + \Delta\mathbf{c}$ with $\Phi(\mathbf{p}, \hat{\mathbf{c}} + \Delta\mathbf{c}, \hat{\mathbf{z}})$. As a result, the transferred shape is a prediction of the outcome of the simulated surgery; it is the zero level set of $\Phi(\mathbf{p}, \hat{\mathbf{c}} + \Delta\mathbf{c}, \hat{\mathbf{z}})$. In more general scenarios, the covariates are unavailable but it is possible to infer them from the measured shapes themselves (see Eqs. 9-10). Therefore, in shape transfer we are not only evolving a shape, but may also first estimate the initial state to be evolved.

## 4 EXPERIMENTS

We evaluate NAISR in terms of shape reconstruction, shape disentanglement, shape evolution, and shape transfer on three datasets: 1) *Starman*, a simulated 2D shape dataset used in (Bône et al., 2020); 2) the ADNI hippocampus 3D shape dataset (Petersen et al., 2010); and 3) a pediatric airway 3D shape dataset. *Starman* serves as the simplest and ideal scenario where sufficient noise-free data for training and evaluating the model is available. While the airway and hippocampus datasets allow for testing on real-world problems of scientific shape analysis, which motivates NAISR.

We can quantitatively evaluate NAISR for shape reconstruction and shape transfer because our dataset contains longitudinal observations. For shape evolution and shape disentanglement, we provide visualizations of shape extrapolations in covariate space to demonstrate that our method can learn a reasonable representation of the deformations governed by the covariates.

Implementation details and ablation studies are available in Section S.3.1 and Section S.3.2 in the supplementary material.

### 4.1 DATASET AND EXPERIMENTAL PROTOCOL

***Starman* Dataset.** This is a synthetic 2D shape dataset obtained from a predefined model as illustrated in Section S.2.1 without additional noise. As shown in Fig. S.4, each starman shape is synthesized by imposing a random deformation representing individual-level variation to the template starman shape. This is followed by a covariate-controlled deformation to the individualized starman shape, representing different poses produced by a starman. 5041 shapes from 1000 starmen are synthesized as the training set; 4966 shapes from another 1000 starmen are synthesized as a testing set.

**ADNI Hippocampus Dataset.** These hippocampus shapes were obtained from the Alzheimer's Disease Neuroimaging Initiative (ADNI) database. The dataset consists of 1632 hippocampus segmentations from magnetic resonance (MR) images. We use an 80%-20% train-test split by patient (instead of shapes); i.e., a given patient cannot simultaneously be in the train and the test set, and therefore no information can leak between these two sets. As a result, the training set consists of 1297

| | Starman | | | | | | ADNI Hippocampus | | | | | | Pediatric Airway | | | | | |
|---|---|---|---|---|---|---|---|---|---|---|---|---|---|---|---|---|---|---|
| | CD↓ | | EMD↓ | | HD↓ | | CD↓ | | EMD↓ | | HD↓ | | CD↓ | | EMD↓ | | HD↓ | |
| | $\mu$ | M | $\mu$ | M | $\mu$ | M | $\mu$ | M | $\mu$ | M | $\mu$ | M | $\mu$ | M | $\mu$ | M | $\mu$ | M |
| DeepSDF | 0.117 | 0.105 | 1.941 | 1.887 | 6.482 | 6.271 | 0.157 | **0.140** | 2.098 | **2.035** | 9.762 | 9.276 | **0.077** | 0.052 | 1.401 | 1.266 | 10.765 | 9.446 |
| A-SDF | 0.173 | 0.092 | 2.010 | 1.668 | 8.806 | 6.949 | 1.094 | 1.162 | 7.156 | 7.667 | 25.092 | 25.938 | 2.647 | 1.178 | 10.302 | 9.068 | 47.172 | 37.835 |
| A-SDF (c) | **0.049** | **0.043** | 1.298 | **1.261** | 5.388 | **4.964** | 0.311 | 0.294 | 3.136 | 3.099 | 13.852 | 13.003 | 0.852 | 0.226 | 4.090 | 2.890 | 30.848 | 21.965 |
| DIT | 0.281 | 0.181 | 2.727 | 2.497 | 10.295 | 8.442 | **0.156** | 0.142 | **2.096** | 2.054 | **9.465** | **9.123** | 0.094 | 0.049 | 1.414 | 1.262 | 11.524 | 10.228 |
| NDF | 1.086 | 0.736 | 5.364 | 4.821 | 21.098 | 19.705 | 0.253 | 0.213 | 2.699 | 2.580 | 11.328 | 10.947 | 0.238 | 0.117 | 2.174 | 1.737 | 14.950 | 12.516 |
| Ours | 0.111 | 0.072 | 1.709 | 1.515 | 7.951 | 7.141 | 0.174 | 0.153 | 2.258 | 2.191 | 10.019 | 9.521 | **0.067** | **0.039** | **1.233** | **1.132** | **10.333** | **8.404** |
| Ours (c) | **0.049** | **0.036** | **1.276** | **1.156** | **5.051** | **4.666** | **0.126** | **0.116** | **1.847** | **1.810** | **8.586** | **8.153** | 0.084 | 0.044 | 1.345 | 1.190 | 10.719 | 8.577 |

Table 2: Quantitative evaluation of shape reconstruction. CD = Chamfer distance. EMD = Earth mover's distance. HD = Hausdorff distance. All metrics are multiplied by $10^2$. **Bold red values** indicate the best scores across all methods. **Bold black values** indicate the 2nd best scores of all methods. **Ours** means the covariates were inferred during testing (see Equation 9). **Ours(c)** means the covariates are used as input during inference (see Equation 10). $\mu$ and M indicate the mean and median of the measurements on the testing shapes respectively. `NAISR` performs well for all three reconstruction tasks while allowing for interpretability.

shapes while the testing set contains 335 shapes. Each shape is associated with 4 covariates (age, sex, AD, education length). AD is a binary variable indicating whether a person has Alzheimer disease.

**Pediatric Airway Dataset.** This dataset contains 357 upper airway shapes to evaluate our method. These shapes are obtained from automatic airway segmentations of computed tomography (CT) images of children with a radiographically normal airway. These 357 airway shape are from 263 patients, 34 of whom have longitudinal observations and 229 of whom have only been observed once. We use a 80%-20% train-test split by patient (instead of shapes). Each shape has 3 covariates (age, weight, sex).

More details, including demographic information, visualizations, and preprocessing steps of the datasets are available in Section S.2 in the supplementary material.

**Comparison Methods.** For shape reconstruction of unseen shapes, we compare our method on the test set with DeepSDF (Park et al., 2019), A-SDF (Mu et al., 2021), DIT (Zheng et al., 2021), and NDF (Sun et al., 2022a). For shape transfer, we compare our method with A-SDF (Mu et al., 2021) because other comparison methods cannot model covariates as summarized in Table 1.

**Metrics.** For evaluation, all target shapes and reconstructed meshes are normalized to a unit sphere (i.e., centered at the origin and uniformly scaled so that the furthest point is at unit distance from the origin) to assure that large shapes and small shapes contribute equally to error measurements. We use the Hausdorff distance, Chamfer distance, and earth mover's distance to evaluate the performance of our shape reconstructions. For shape transfer, considering that a perfectly consistent image acquisition process is impossible for different observations (e.g., head positioning might slightly vary across timepoints for the airway data), we visualize the transferred shapes and evaluate based on the difference between the *volumes* of the reconstructed shapes and the target shapes on the hippocampus and airway dataset.

### 4.2 SHAPE RECONSTRUCTION

The goal of our shape reconstruction experiment is to demonstrate that `NAISR` can provide competitive reconstruction performance while providing interpretability. Table 2 shows the quantitative evaluations and demonstrates the excellent reconstruction performance of `NAISR`. Figure 2 visualizes reconstructed shapes. We observe that implicit shape representations can complete missing shape parts which can benefit further shape analysis. A-SDF (Mu et al., 2021) works well for representing *Starman* shapes but cannot reconstruct real 3D medical shapes successfully. The reason might be the time span of our longitudinal data for each patient is far shorter than the time span across the entire dataset, mixing individual differences and covariate-controlling differences. A-SDF may fail to disentangle such mixed effects (from individuals and covariates), but instead memorizes training shapes by their covariates **c**. In contrast, the additive architecture of `NAISR` prevents the model from memorizing training shapes through covariates **c**. More discussions are available in Section S.3.3 of the supplementary material.

### 4.3 SHAPE TRANSFER

Table 4 shows an airway shape transfer example for a cancer patient who was scanned 11 times. We observe that our method can produce complete transferred shapes that correspond well with the measured shapes. Table 3 shows quantitative results for the volume differences between our transferred shapes and the gold standard shapes. Our method performs best on the real datasets

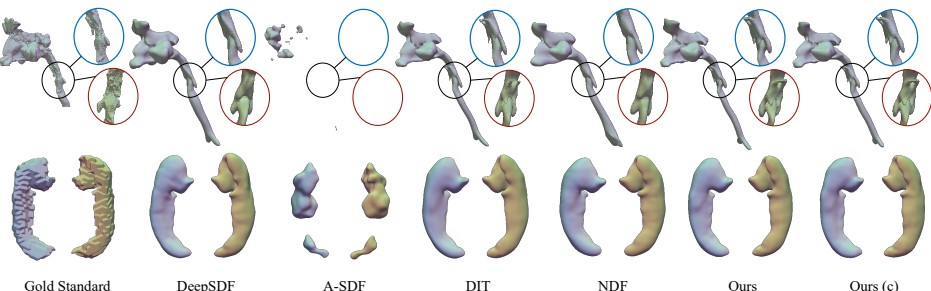

Gold Standard    DeepSDF    A-SDF    DIT    NDF    Ours    Ours (c)

Figure 2: Visualizations of airway and hippocampus reconstruction with different methods. The red and blue circles show the structure in the black circle from two different views. Hippocampus shapes are plotted with two $180°$-flipped views. NAISR can produce detailed and accurate reconstructions as well as impute missing airway parts. More visualizations are available in Section S.3.3 of the supplementary material.

| | | Starman | | | | | | ADNI Hippocampus | | Pediatric Airway | |
|---|---|---|---|---|---|---|---|---|---|---|---|
| | | CD↓ | | EMD↓ | | HD↓ | | VD↓ | | VD↓ | |
| | w.C. | $\mu$ | M | $\mu$ | M | $\mu$ | M | $\mu$ | M | $\mu$ | M |
| A-SDF | ✗ | **0** | **0** | **0.009** | **0.008** | **0.036** | **0.034** | 0.518 | 0.488 | 81.074 | 82.918 |
| A-SDF | ✓ | **0** | **0** | **0.009** | **0.009** | **0.036** | **0.035** | 0.215 | 0.177 | 41.460 | 40.956 |
| Ours | ✗ | 0.003 | 0.002 | 0.025 | 0.023 | 0.094 | 0.077 | **0.086** | **0.063** | 12.820 | **8.837** |
| Ours | ✓ | 0.009 | 0.002 | 0.031 | 0.025 | 0.116 | 0.083 | **0.089** | **0.071** | **11.227** | 9.653 |

Table 3: Quantitative evaluation of shape transfer. Statistics of Volume Difference (VD, $cm^3$) between transferred shapes and gold standard shapes. w.C. abbreviates *with covariates*. A ✓ in w.C. indicates the inference follows Equation 10. A ✗ in w.C. indicates the inference follows Equation 9. **Red bold scores** indicate the best performance across all methods and **bold scores** indicate the 2nd best. NAISR results in significantly improved volume estimates for real medical shapes.

while A-SDF (the only other model supporting shape transfer) works slightly better on the synthetic *Starman* dataset. Our results demonstrate that NAISR is capable of transferring shapes to other timepoints $\mathcal{S}^t$ from a given initial state $\mathcal{S}^0$.

| #time | 0 | 1 | 2 | 3 | 4 | 5 | 6 | 7 | 8 | 9 | 10 |
|---|---|---|---|---|---|---|---|---|---|---|---|

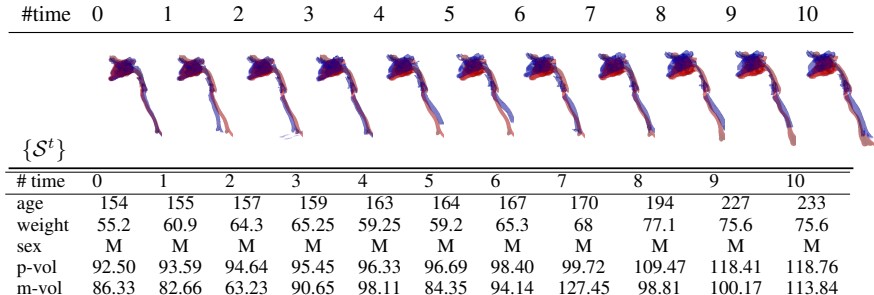

| $\{\mathcal{S}^t\}$ | | | | | | | | | | | |
|---|---|---|---|---|---|---|---|---|---|---|---|
| # time | 0 | 1 | 2 | 3 | 4 | 5 | 6 | 7 | 8 | 9 | 10 |
| age | 154 | 155 | 157 | 159 | 163 | 164 | 167 | 170 | 194 | 227 | 233 |
| weight | 55.2 | 60.9 | 64.3 | 65.25 | 59.25 | 59.2 | 65.3 | 68 | 77.1 | 75.6 | 75.6 |
| sex | M | M | M | M | M | M | M | M | M | M | M |
| p-vol | 92.50 | 93.59 | 94.64 | 95.45 | 96.33 | 96.69 | 98.40 | 99.72 | 109.47 | 118.41 | 118.76 |
| m-vol | 86.33 | 82.66 | 63.23 | 90.65 | 98.11 | 84.35 | 94.14 | 127.45 | 98.81 | 100.17 | 113.84 |

Table 4: Airway shape transfer for a patient. Blue: gold standard shapes; red: transferred shapes with NAISR. The table below lists the covariates (age/month, weight/kg, sex) for the shapes above. P-vol(predicted volume) is the volume ($cm^3$) of the transferred shape by NAISR covariates following Equation 9. M-vol (measured volume) is the volume ($cm^3$) of the shapes based on the actual imaging. NAISR can capture the trend of growing volume with age and weight while producing clear, complete, and topology-consistent shapes. Note that measured volumes may differ depending on the CT imaging field of view. More visualizations are available in Section S.3.4 in the supplementary material.

## 4.4 SHAPE DISENTANGLEMENT AND EVOLUTION

Figure 3 shows that NAISR is able to extrapolate reasonable shape changes when varying covariates. These results illustrate the capabilities of NAISR for shape disentanglement and to capture shape evolutions. A-SDF and NAISR both produce high-quality *Starman* shapes because the longitudinal data is sufficient for the model to discover the covariate space. However, for real data, only NAISR can produce realistic 3D hippocampi and airways reflecting the covariates' influences on template shapes. Note that when evolving shapes along a single covariate $c_i$, the deformations from other covariates are naturally set to **0** by our model construction (see Section 3.2). As a result, the shapes in the yellow and green boxes in Figure 3 represent the disentangled shape evolutions along different covariates respectively. The shapes in the other grid positions can be extrapolated using $\Phi(\cdot)$. By inspecting the volume changes in the covariate space in Figure 3, we observe that age is more important for airway volume than weight, and Alzheimer disease influences hippocampal volume. These observations from our generated shapes are consistent with clinical expectations (Luscan et al., 2020; Gosche et al., 2002), suggesting that NAISR is able to extract hidden knowledge from data and is able to generate interpretable results directly as 3D shapes.

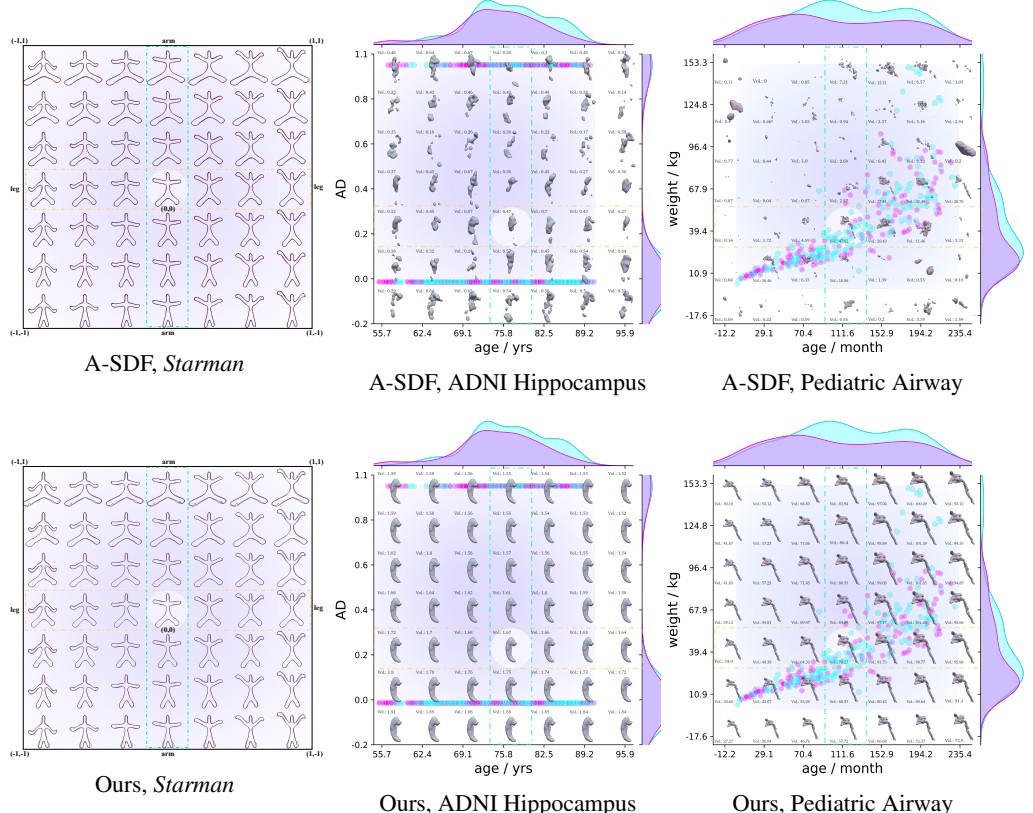

Figure 3: Template shape extrapolation in covariate space using A-SDF and NAISR on three datasets. For the *Starman* shape extrapolations, the blue shapes are the groundtruth shapes and the red shapes are the reconstructions. The shapes in the middle white circles are the template shapes. The template shape is generated with zero latent code and is used to create a template covariate space. The shapes in the green and yellow boxes are plotted with $\{\Phi_i\}$, representing the disentangled shape evolutions along each covariate respectively. The purple shadows over the space indicate the covariate range that the dataset covers. Cyan points represent male and purple points female patients in the dataset. The points represent the covariates of all patients in the dataset. The colored shades at the boundary represent the covariate distributions stratified by sex. Example 3D shapes in the covariate space are visualized with their volumes ($cm^3$) below. NAISR is able to extrapolate the shapes in the covariate space given either an individualized latent code $\mathbf{z}$ or template latent code $\mathbf{0}$, whereas A-SDF struggles. The supplementary material provides more visualizations of individualized covariate spaces in Section S.3.5. (Best viewed zoomed.)

## 5  LIMITATIONS AND FUTURE WORK

Invertible transforms are often desirable for shape correspondences but not guaranteed in NAISR. Invertibility could be guaranteed by representing deformations via velocity fields, but such parameterizations are costly because of the required numerical integration. In future work, we will develop efficient invertible representations, which will ensure topology preservation. So far we only indirectly assess our model by shape reconstruction and transfer performance. Going forward we will include patients with airway abnormalities. This will allow us to explore if our estimated model of normal airway shape can be used to detect airway abnormalities. Introducing group sparsity (Yin et al., 2012; Chen et al., 2017) to NAISR for high-dimensional covariates is also promising future work.

## 6  CONCLUSION

We proposed NAISR, a 3D neural additive model for interpretable shape representation. We tested NAISR on three different datasets and observed particularly good performance on real 3D medical datasets. Compared to other shape representation methods, NAISR 1) captures the effect of individual covariates on shapes; 2) can transfer shapes to new covariates, e.g., to infer anatomy development; and 3) can provide shapes based on extrapolated covariates. NAISR is the first approach combining deep implicit shape representations based on template deformation with the ability to account for covariates. We believe our work is an exciting start for a new line of research: interpretable neural shape models for scientific discovery.

ACKNOWLEDGEMENT

The research reported in this publication was supported by NIH grant 1R01HL154429. The content is solely the responsibility of the authors and does not necessarily represent the official views of the NIH. Data collection and sharing for this project was funded by the Alzheimer's Disease Neuroimaging Initiative (ADNI) (National Institutes of Health Grant U01 AG024904) and DOD ADNI (Department of Defense award number W81XWH-12-2-0012). ADNI is funded by the National Institute on Aging, the National Institute of Biomedical Imaging and Bioengineering, and through generous contributions from the following: AbbVie, Alzheimer's Association; Alzheimer's Drug Discovery Foundation; Araclon Biotech; BioClinica, Inc.; Biogen; Bristol-Myers Squibb Company; CereSpir, Inc.; Cogstate; Eisai Inc.; Elan Pharmaceuticals, Inc.; Eli Lilly and Company; EuroImmun; F. Hoffmann-La Roche Ltd and its affiliated company Genentech, Inc.; Fujirebio; GE Healthcare; IXICO Ltd.;Janssen Alzheimer Immunotherapy Research & Development, LLC.; Johnson & Johnson Pharmaceutical Research & Development LLC.; Lumosity; Lundbeck; Merck & Co., Inc.;Meso Scale Diagnostics, LLC.; NeuroRx Research; Neurotrack Technologies; Novartis Pharmaceuticals Corporation; Pfizer Inc.; Piramal Imaging; Servier; Takeda Pharmaceutical Company; and Transition Therapeutics. The Canadian Institutes of Health Research is providing funds to support ADNI clinical sites in Canada. Private sector contributions are facilitated by the Foundation for the National Institutes of Health (www.fnih.org). The grantee organization is the Northern California Institute for Research and Education, and the study is coordinated by the Alzheimer's Therapeutic Research Institute at the University of Southern California. ADNI data are disseminated by the Laboratory for Neuro Imaging at the University of Southern California.

## REPRODUCIBILITY STATEMENT

We are dedicated to ensuring the reproducibility of `NAISR` to facilitate more scientific discoveries on shapes. To assist researchers in replicating and building upon our work, we made the following efforts.

- **Model & Algorithm**: Our paper provides detailed descriptions of the model architectures (see Section 3), implementation details (see Section S.3.1), and ablation studies (see Section S.3.2). We have submitted our source code. The implementation of `NAISR` will be made publicly available.

- **Datasets & Experiments**: We provide extensive illustrations and visualizations for the datasets we used. To ensure transparency and ease of replication, the exact data processing steps, from raw data to processed input, are outlined in Section S.2 of the supplementary materials. We expect our detailed supplementary material to ensure the reproducibility of our method and the understandability of our experimental results. We also have submitted the code for synthesizing the 2D *Starman* dataset so that researchers can easily reproduce the results.

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

## SUPPLEMENTARY MATERIAL FOR NAISR

### S.1 RELATED WORK

**Deep Implicit Functions.** Compared with geometry representations such as voxel grids (Wu et al., 2016; 2015; 2018), point clouds (Achlioptas et al., 2018; Yang et al., 2018; Zamorski et al., 2020) and meshes (Groueix et al., 2018; Wen et al., 2019; Zhu et al., 2019), DIFs are able to capture highly detailed and complex 3D shapes using a relatively small amount of data (Park et al., 2019; Mu et al., 2021; Zheng et al., 2021; Sun et al., 2022a; Deng et al., 2021). They are based on classical ideas of level set representations (Sethian, 1999; Osher & Fedkiw, 2005); however, whereas these classical level set methods represent the level set function on a grid, DIFs parameterize it as a *continuous function*, e.g., by a neural network. Hence, DIFs are not reliant on meshes, grids, or a discrete set of points. This allows them to efficiently represent natural-looking surfaces (Gropp et al., 2020; Sitzmann et al., 2020; Niemeyer et al., 2019). Further, DIFs can be trained on a diverse range of data (e.g., shapes and images), making them more versatile than other shape representation methods. This makes them useful in applications ranging from computer graphics, to virtual reality, and robotics (Gao et al., 2022; Yang et al., 2022a; Phongthawee et al., 2022; Shen et al., 2021). *We therefore formulate NAISR based on DIFs.*

**Neural Deformable Models** Neural Deformable Models (NDMs) establish point correspondences with DIFs. In computer graphics, there has been increasing interest in NDMs to animate or edit scenes (Liu et al., 2022; Park et al., 2021a; Bao et al., 2023; Zheng et al., 2023), objects (Duggal & Pathak, 2022; Lei & Daniilidis, 2022; Bao et al., 2023; Niemeyer et al., 2019; Zhang et al., 2023; Shuai et al., 2023), and digital humans (Liu et al., 2021; Chen et al., 2021; Niemeyer et al., 2019; Park et al., 2021b;a; Peng et al., 2021; Zhang & Chen, 2022; Zhang et al., 2023).

Establishing point correspondences is also important to help experts to detect, understand, diagnose, and track diseases. NDMs have shown to be effective in exploring point correspondences for medical images (Han et al., 2023b; Tian et al., 2023; Sun et al., 2022b; Wolterink et al., 2022a; Zou et al., 2023) and shapes (Sun et al., 2022a; Yang et al., 2022b; Han et al., 2023a)

Among NDMs for shape representations, ImplicitAtlas (Yang et al., 2022b), DIF-Net (Deng et al., 2021), DIT (Zheng et al., 2021), and NDF (Sun et al., 2022a) were proposed to capture point correspondence between target and template shapes within NDMs. Currently, no continuous deformable shape representation which models the effects of covariates exists. *NAISR provides a model with such capabilities.*

**Disentangled Representation Learning.** Disentangled representation learning (DRL) has been explored in a variety of domains, including computer vision (Shoshan et al., 2021; Ding et al., 2020; Zhang et al., 2018b;a; Xu et al., 2021; Yang et al., 2020), natural language processing (John et al., 2018), and medical image analysis (Chartsias et al., 2019; Bercea et al., 2022).

For example, Wei et al. built a mesh editing model based on disentangled semantic parameters. Their model learns from simulated datasets which are based on parameterized models and pre-defined templates (Wei et al., 2020). DRL has also emerged in the context of implicit representations as a promising approach for 3D computer vision. By disentangling the underlying factors of variation, such as object shape, orientation, and texture, DRL can facilitate more effective 3D object recognition, reconstruction, and manipulation (Stammer et al., 2022; Zhang et al., 2018b;a; Xu et al., 2021; Yang et al., 2020; 2022a; Gao et al., 2022; Tewari et al., 2022).

Besides DRL in computer vision, medical data is typically associated with various covariates which should be taken into account during analyses. Taking (Chu et al., 2022) as an example, when observing a tumor's progression, it is difficult to know whether the variation of a tumor's progression is due to time-varying covariates or due to treatment effects. Therefore, being able to disentangle different effects is highly useful for a representation to promote understanding and to be able to quantify the effect of covariates on observations. *NAISR provides a disentangled representation and allows us to capture the shape effects of covariates.*

**Articulated Shapes.** There is significant research focusing on articulated shapes, mostly on humans (Palafox et al., 2021; Chen et al., 2021; Tretschk et al., 2020; Deng et al., 2020). There is also a

line of work on articulated general objects, e.g., A-SDF (Mu et al., 2021), NASAM (Wei et al., 2022) and LEPARD (Liu et al., 2023). A-SDF (Mu et al., 2021) uses articulation as an additional input to control generated shapes, while NASAM (Wei et al., 2022) and LEPARD (Liu et al., 2023) learns the latent space of articulation without articulation as supervision.

The aforementioned works on articulated objects assume that each articulation affects a separate object part. This is easy to observe, e.g., the angles of the two legs of a pair of eyeglasses. Hence, although A-SDF (Mu et al., 2021) and 3DAttriFlow (Wen et al., 2022) can disentangle articulations from geometry, they do not disentangle different covariates and their disentanglements are not composable. However, in medical scenarios, covariates often affect shapes in a more entangled and complex way, for example, a shape might simultaneously be influenced by sex, age, and weight. Dalca et al. (Dalca et al., 2019) use templates conditioned on covariates for image registration. However, they did not explore covariate-specific deformations, shape representations or shape transfer. *NAISR allows us to account for such complex covariate interactions.*

**Explainable Artificial Intelligence.** The goal of eXplainable Artificial Intelligence (XAI) is to provide human-understandable explanations for decisions and actions of an AI model. Various flavors of XAI exist, including counterfactual inference (Berrevoets et al., 2021; Moraffah et al., 2020; Thiagarajan et al., 2020; Chen et al., 2022), attention maps (Zhou et al., 2016; Jung & Oh, 2021; Woo et al., 2018), feature importance (Arik & Pfister, 2021; Ribeiro et al., 2016; Agarwal et al., 2020), and instance retrieval (Crabbe et al., 2021). NAISR is inspired by neural additive models (NAMs) (Agarwal et al., 2020) which in turn are inspired by generalized additive models (GAMs) (Hastie, 2017). NAMs are based on a linear combination of neural networks each attending to a single input feature. NAISR extends this concept to interpretable 3D shape representations. This is significantly more involved as, unlike for NAMs and GAMs, we are no longer dealing with scalar values, but with 3D shapes. *NAISR provides interpretable results by capturing spatial deformations with respect to an estimated atlas shape such that individual covariate effects can be distinguished.*

## S.2 DATASET

### S.2.1 STARMAN DATASET

Figure S.4 illustrates how each sample in the dataset is simulated. 5041 shapes from 1000 different starmen are simulated as the training set. 4966 shapes from another 1000 starmen are simulated as a testing set. The number of movements for each individual comes from a uniform distribution $\mathcal{U}_{\{1,...10\}}$.

The deformation for arms can be represented as

$$\mathbf{d}_i(\mathbf{p}) = \alpha \cdot exp(-\frac{(\mathbf{p} - \mathbf{c}_i)^T(\mathbf{p} - \mathbf{c}_i)}{2\sigma^2}) \cdot \mathbf{m}_0 \,, i = 0, 1; \sigma = 0.5 \,. \tag{S.12}$$

The deformation for legs can be represented as

$$\mathbf{d}_i(\mathbf{p}) = \beta \cdot exp(-\frac{(\mathbf{p} - \mathbf{c}_i)^T(\mathbf{p} - \mathbf{c}_i)}{2\sigma^2}) \cdot \mathbf{m}_1 \,, i = 2, 3; \sigma = 0.5 \,. \tag{S.13}$$

We sample the covariates $\alpha$ and $\beta$ from a uniform distribution $\mathcal{U}_{[-1,1]}$. The overall deformation $\mathbf{D}$ is the sum of the covariates-controlling deformations $\{\mathbf{d}_i\}$ imposed on the individual starman shape, as

$$\mathbf{D} = \Sigma_i \mathbf{d}_i(\mathbf{d}_r(\mathbf{p}) + \mathbf{p}) \,. \tag{S.14}$$

### S.2.2 ADNI HIPPOCAMPUS

The ADNI hippocampus dataset [‡] consists of 1632 hippocampus segmentations from magnetic resonance (MR) images from the ADNI dataset, 80% (1297 shapes) of which are used for training

---

[‡]Data used in the preparation of this article were obtained from the Alzheimer's Disease Neuroimaging Initiative (ADNI) database (adni.loni.usc.edu). The ADNI was launched in 2003 as a public-private

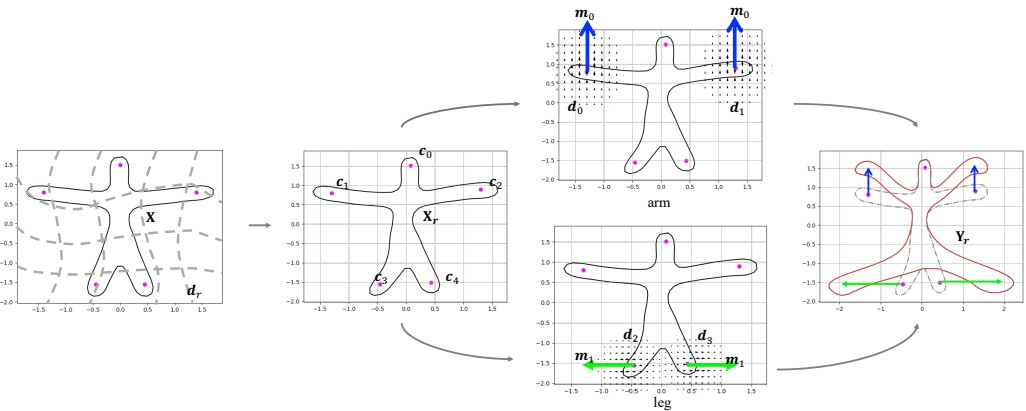

Figure S.4: Visualization of the *Starman* dataset simulation. The template shape $\mathbf{X}$ is in solid black, the control points $\{\mathbf{c}_i\}$ are the five points in magenta. The grey dashed lines represent the individual random deformation $\mathbf{d}_r$ to the template and those control points $\{\mathbf{c}_i\}$, yielding an individualized template $\mathbf{X}_r$. The moving direction $\mathbf{m}_0$ controlling the two arms is shown as a **bold blue arrow**. The moving direction $\mathbf{m}_1$ controlling the two legs is shown as a **bold green arrow**. The velocity fields $\mathbf{d}_0$ and $\mathbf{d}_1$ control the upward/downward movement of two arms correspondingly. The velocity fields $\mathbf{d}_2$ and $\mathbf{d}_3$ controls the splits of two legs correspondingly. The individualized template shape $\mathbf{X}_r$ is deformed by $\mathbf{d} = \Sigma_i \mathbf{d}_i$ to $\mathbf{Y}_r$ (the red shape), representing a person moving their arms and legs. The covariates $\alpha$ and $\beta$ decide how much the arm is lifted and how much the legs are split.

and 20% (335 shapes) for testing. Each shape is associated with 4 covariates (age, sex, AD, education length). AD is a binary variable that represents whether a person has Alzheimer disease. AD=1 indicates a person has Alzheimer disease. Table S.5 shows the distribution of the number of observations across patients. Table S.6 shows the hippocampus shapes and the demographic information of an example patient. Table S.7 shows the shapes and demographic information at different age percentiles for the whole data set. We observe that the time span of our longitudinal data for each patient is far shorter than the time span across the entire dataset, indicating the challenge of capturing spatiotemporal dependencies over large time spans between shapes while accounting for individual differences between patients.

| # observations | 1 | 2 | 3 | 4 | 5 | 6 |
|---|---|---|---|---|---|---|
| # patients | 3 | 10 | 410 | 5 | 7 | 54 |

Table S.5: Number of patients for a given number of observations for the ADNI dataset. For example, the 1st column indicates that there are 3 patients who were only observed once.

### S.2.3 PEDIATRIC AIRWAY

The airway shapes are extracted from computed tomography (CT) images. We use real CT images of children ranging in age from 1 month to ∼19 years old. Acquiring CT images is costly. Further, CT uses ionizing radiation which should be avoided, especially in children, due to cancer risks. Hence, it is difficult to acquire such CTs for many children. Instead, our data was acquired by serendipity from children who received CTs for reasons other than airway obstructions (e.g., because they had cancer). This also explains why it is difficult to acquire longitudinal data. E.g., one of our patients has 11 timepoints because a very sick child had to be scanned 11 times. *Note that our data is very different from typical CV datasets which can be more readily acquired at scale or may even already*

partnership, led by Principal Investigator Michael W. Weiner, MD. The primary goal of ADNI has been to test whether serial magnetic resonance imaging (MRI), positron emission tomography (PET), other biological markers, and clinical and neuropsychological assessment can be combined to measure the progression of mild cognitive impairment (MCI) and early Alzheimer's disease (AD). For up-to-date information, see `www.adni-info.org`.

| # time | 0 | 1 | 2 | 3 | 4 | 5 |
|---|---|---|---|---|---|---|

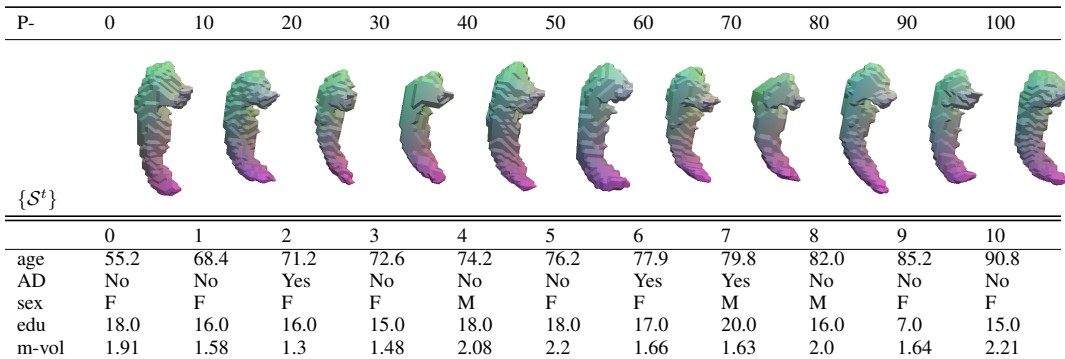

| $\{\mathcal{S}^t\}$ | 0 | 1 | 2 | 3 | 4 | 5 |
|---|---|---|---|---|---|---|
| age | 75.6 | 75.7 | 76.2 | 76.2 | 76.7 | 76.7 |
| AD | No | No | No | No | No | No |
| sex | F | F | F | F | F | F |
| edu | 20.0 | 20.0 | 20.0 | 20.0 | 20.0 | 20.0 |
| m-vol | 2.26 | 2.38 | 2.2 | 2.35 | 2.27 | 2.16 |

Table S.6: Visualization and demographic information of observations of a patient in the ADNI hippocampus dataset. Shapes are plotted with their covariates (age/yrs, AD, sex, edu(education length)/yrs) printed in the table. M-vol (measured volume) is the volume ($cm^3$) of the gold standard shapes based on the actual imaging.

| P- | 0 | 10 | 20 | 30 | 40 | 50 | 60 | 70 | 80 | 90 | 100 |
|---|---|---|---|---|---|---|---|---|---|---|---|

| $\{\mathcal{S}^t\}$ | 0 | 1 | 2 | 3 | 4 | 5 | 6 | 7 | 8 | 9 | 10 |
|---|---|---|---|---|---|---|---|---|---|---|---|
| age | 55.2 | 68.4 | 71.2 | 72.6 | 74.2 | 76.2 | 77.9 | 79.8 | 82.0 | 85.2 | 90.8 |
| AD | No | No | Yes | No | No | No | Yes | Yes | No | No | No |
| sex | F | F | F | F | M | F | F | M | M | F | F |
| edu | 18.0 | 16.0 | 16.0 | 15.0 | 18.0 | 18.0 | 17.0 | 20.0 | 16.0 | 7.0 | 15.0 |
| m-vol | 1.91 | 1.58 | 1.3 | 1.48 | 2.08 | 2.2 | 1.66 | 1.63 | 2.0 | 1.64 | 2.21 |

Table S.7: Visualization and demographic information of our ADNI Hippocampus 3D shape dataset. Shapes of $\{0, 10, 20, 30, 40, 50, 60, 70, 80, 90, 100\}$-th age percentiles are plotted with their covariates (age/yrs, AD, sex, edu(education length)/yrs) printed in the table. M-vol (measured volume) is the volume ($cm^3$) of the gold standard shapes based on the actual imaging.

*exist based on internet photo collections. This is impossible for our task because image acquisition risks always have to be justified by patient benefits.*

Our dataset includes 229 cross-sectional observations (where a patient was only imaged once) and 34 longitudinal observations. Each shape has 3 covariates (age, weight, sex) and 11 annotated anatomical landmarks. Errors in the shapes $\{\mathcal{S}^k\}$ may arise from image segmentation error, differences in head positioning, missing parts of the airway shapes due to incomplete image coverage, and dynamic airway deformations due to breathing. Table S.8 shows the distribution of the number of observations across patients. Most of the patients in the dataset only have one observation; only 22 patients have $\geq 3$ observation times. Table S.9 shows the airway shapes and the demographic information of an example patient. Table S.10 shows the shapes and demographic information at different age percentiles for the whole data set. Similar to the ADNI hippocampus dataset, the time span of the longitudinal data for each patient is far shorter than the time span across the entire dataset, which poses a significant shape analysis challenge for realistic medical shapes.

| # observations | 1 | 2 | 3 | 4 | 5 | 6 | 7 | 9 | 11 |
|---|---|---|---|---|---|---|---|---|---|
| # patients | 229 | 12 | 6 | 8 | 3 | 2 | 1 | 1 | 1 |

Table S.8: Number of patients for a given number of observations for the pediatric airway dataset. For example, the 1st column indicates that there are 229 patients who were only observed once.

**Data Processing.** For the ADNI hippocampus dataset and the pediatric airway dataset, the shape meshes are extracted using Marching Cubes (Lorensen & Cline, 1987; Van der Walt et al., 2014) to obtain coordinates and normal vectors of on-surface points. The hippocapus shapes are rigidly aligned using the ICP algorithm (Arun et al., 1987). The airway shapes are rigidly aligned using the anatomical landmarks. The true vocal cords landmark is set to the origin. We follow the implementation in (Park et al., 2019) to sample 500,000 off-surface points. During training, it is important to preserve the scale information. We therefore scale all meshes with the same constant.

## S.3 EXPERIMENTS

Section S.3.1 describes implementation details Section S.3.2 describe the ablation study. Section S.3.3, Section S.3.4, and Section S.3.5 show additional experimental results for shape reconstruction, shape transfer, and disentangled shape evolution, respectively.

### S.3.1 IMPLEMENTATION DETAILS

Each subnetwork, including the template network $\mathcal{T}$ and the displacement networks $\{f_i\}$, are all parameterized with an $N_l$-layer MLP using *sine* activations. We use $N_l$=8 for `Starman` and the

| #time | 0 | 1 | 2 | 3 | 4 | 5 | 6 | 7 | 8 |
|---|---|---|---|---|---|---|---|---|---|

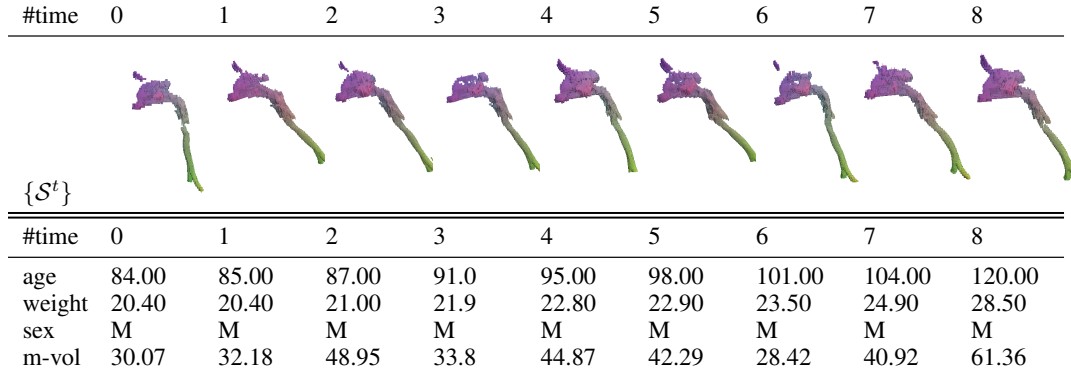

| $\{\mathcal{S}^t\}$ | | | | | | | | | |
|---|---|---|---|---|---|---|---|---|---|
| #time | 0 | 1 | 2 | 3 | 4 | 5 | 6 | 7 | 8 |
| age | 84.00 | 85.00 | 87.00 | 91.0 | 95.00 | 98.00 | 101.00 | 104.00 | 120.00 |
| weight | 20.40 | 20.40 | 21.00 | 21.9 | 22.80 | 22.90 | 23.50 | 24.90 | 28.50 |
| sex | M | M | M | M | M | M | M | M | M |
| m-vol | 30.07 | 32.18 | 48.95 | 33.8 | 44.87 | 42.29 | 28.42 | 40.92 | 61.36 |

Table S.9: Visualization and demographic information of observations of a patient in our 3D airway shape dataset. Shapes are plotted with their covariates (age/month, weight/kg, sex) printed in the table. M-vol (measured volume) is the volume ($cm^3$) of the gold standard shapes based on the actual imaging.

| P- | 0 | 10 | 20 | 30 | 40 | 50 | 60 | 70 | 80 | 90 | 100 |
|----|---|----|----|----|----|----|----|----|----|----|-----|

$\{S^t\}$

| P- | 0 | 10 | 20 | 30 | 40 | 50 | 60 | 70 | 80 | 90 | 100 |
|----|---|----|----|----|----|----|----|----|----|----|-----|
| age | 1.00 | 23.00 | 55.00 | 71.00 | 89.00 | 111.00 | 129.00 | 161.00 | 179.00 | 199.00 | 233.00 |
| weight | 3.90 | 14.20 | 20.10 | 21.80 | 19.70 | 32.85 | 44.80 | 21.30 | 59.00 | 93.90 | 75.60 |
| sex | M | M | F | F | M | M | M | F | F | F | M |
| m-vol | 4.56 | 16.84 | 29.53 | 28.91 | 27.31 | 70.90 | 71.23 | 43.34 | 78.63 | 102.35 | 113.84 |

Table S.10: Visualization and demographic information of our 3D airway shape dataset. Shapes of $\{0, 10, 20, 30, 40, 50, 60, 70, 80, 90, 100\}$-th age percentiles are plotted with their covariates (age/month, weight/kg, sex) printed in the table. M-vol (measured volume) is the volume ($cm^3$) of the gold standard shapes based on the actual imaging.

| Methods | DeepSDF | A-SDF | DIT | NDF | Ours | | |
|---------|---------|-------|-----|-----|------|------|------|
| | | | | | Starman | ADNI Hippocampus | Pediatric Airway |
| #params | 2.24M | 1.98M | 1.92M | 0.34M | 1.33M | 2.26M | 1.26M |

Table S.11: Number of parameters of the different models.

ADNI hippocampus dataset; we use $N_l$=6 for the pediatric airway dataset. There are 256 hidden units in each layer. The architecture of the $\{f_i\}$ follows DeepSDF (Park et al., 2019), in which a skip connection is used to concatenate the input of $(\mathbf{p}, c_i)$ to the input of the middle layer, as shown in Fig. S.5. We use a latent code $\mathbf{z}$ of dimension 256 ($L = 256$).

We follow SIREN (Sitzmann et al., 2020) for the network architecture. SIREN uses periodic activation functions for implicit neural representations and demonstrates that networks which use periodic functions (such as sinusoidal functions) as activations, are well suited for representing complex natural signals and their derivatives. We also follow SIREN's initialization to draw weights according to a uniform distribution $\mathcal{W} \sim \mathcal{U}\left(-\sqrt{\frac{6}{\omega_0^2 D_{in}}}, \sqrt{\frac{6}{\omega_0^2 D_{in}}}\right)$ ($D_{in}$ is the input dimension and $\omega_0$ is the scaling factor of the SIREN layers, which is set to 30).

Table S.11 lists the number of model parameters.

For each training iteration, the number of points sampled from each shape is 750 ($N = 750$), of which 500 are on-surface points ($N_{on} = 500$) and the others are off-surface points ($N_{off} = 250$). We train NAISR for 3000 epochs for the airway dataset and 300 epochs for the ADNI hippocampus and Starman datasets using Adam (Kingma & Ba, 2014) with a learning rate $5e-5$ and batch size of 64. Also, we jointly optimize the latent code $\mathbf{z}$ with NAISR using Adam (Kingma & Ba, 2014) with a learning rate of $1e-3$.

During training, $\lambda_1 = \lambda_5 = 1 \cdot 10$; $\lambda_2 = 3 \cdot 10$; $\lambda_3 = 1 \cdot 10$, $\lambda_4 = 1 \cdot 10^2$. For $\mathcal{L}_{lat}$, $\lambda_6 = \frac{2}{L}$; $\sigma = 0.01$ (following DeepSDF (Park et al., 2019)). During inference, the latent codes are optimized for $N_t$ iterations with a learning rate of $5e-3$. $N_t$ is set to 800 for the pediatric airway dataset; $N_t$ is set to 200 for the *Starman* and ADNI Hippocampus datasets.

**Computational Runtime** The model is trained on an Nvidia GeForce RTX 3090 GPU for approximately 12 hours for 3000 epochs for the airway dataset. For each shape, it takes around 130 seconds for 800 iterations and 30 seconds for 200 iterations to infer the latent code $\mathbf{z}$ and the covariates $\mathbf{c}$ respectively. However, we observed that 200 iterations are sufficient to produce a reasonable reconstruction. Then, it takes approximately 30 seconds to sample the SDF field and apply Marching Cubes to extract the shape mesh. Regarding shape transfer, evolution, and disentanglement for a specific case, we first need to optimize the latent code $\mathbf{z}$ as for the shape reconstruction. Once

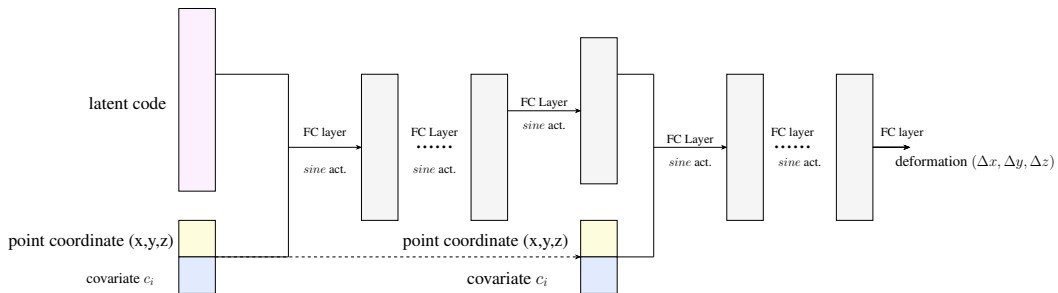

Figure S.5: Construction of $f_i$

| Methods | variants | influenced term | CD ↓ | | EMD ↓ | | HD ↓ | |
|---|---|---|---|---|---|---|---|---|
| | | | $\mu$ | M | $\mu$ | M | $\mu$ | M |
| Ours | $\lambda_1 = \lambda_5 = 0$ | $\mathcal{L}_{Eikonal}$ | 0.072 | 0.047 | 1.447 | 1.323 | 10.426 | 8.716 |
| Ours | $\lambda_4 = 0$ | $\mathcal{L}_{Dirichlet\_off\_surf}$ | 4.323 | 4.481 | 1.374 | 1.244 | 68.527 | 69.715 |
| Ours | $\lambda_3 = 0$ | $\mathcal{L}_{Neumann}$ | 0.081 | 0.051 | 1.449 | 1.307 | 10.269 | 8.546 |
| Ours | $\lambda_2 = 0$ | $\mathcal{L}_{Dirichlet\_on\_surf}$ | 0.124 | 0.077 | 1.912 | 1.682 | 10.803 | 8.916 |
| Ours | $\lambda_6 = 0$ | $\mathcal{L}_{lat}$ | 0.045 | 0.023 | 0.980 | 0.890 | 8.920 | 7.028 |
| Ours | $\lambda_6* = 0.01$ | $\mathcal{L}_{lat}$ | 0.049 | 0.023 | 1.053 | 0.924 | 9.064 | 7.041 |
| Ours | $\lambda_6* = 0.1$ | $\mathcal{L}_{lat}$ | 0.056 | 0.031 | 1.126 | 1.015 | 9.578 | 7.831 |
| Ours | $\lambda_6* = 1$ | $\mathcal{L}_{lat}$ | 0.067 | 0.039 | 1.251 | 1.143 | 10.333 | 8.404 |
| Ours | $\lambda_6* = 10$ | $\mathcal{L}_{lat}$ | 0.075 | 0.049 | 1.368 | 1.270 | 11.111 | 9.176 |
| Ours | $\lambda_6* = 100$ | $\mathcal{L}_{lat}$ | 0.100 | 0.073 | 1.607 | 1.532 | 13.456 | 11.853 |

Table S.12: Ablation study: quantitative evaluation of shape reconstruction. Ours means covariates are not used as additional input to `NAISR`. The shadowed line is what we report in the main text.

the latent code **z** is optimized or assigned (e.g., to **0** as template), it will take around 30 seconds to produce a new mesh controlled by the covariate **c**.

**Comparison Methods.** For shape reconstruction of unseen shapes, we compare our method on the test set with DeepSDF (Park et al., 2019) A-SDF (Mu et al., 2021) DIT (Zheng et al., 2021) and NDF (Sun et al., 2022a). For shape transfer, we compare our method with A-SDF (Mu et al., 2021) because other comparison methods cannot model covariates as summarized in Table 1. The original implementations of the comparison methods did not produce satisfying reconstructions on our dataset. We therefore improved them by using our reconstruction losses and by using the `SIREN` backbone (Sitzmann et al., 2020) in DeepSDF (Park et al., 2019), A-SDF (Mu et al., 2021), and the template networks in DIT (Zheng et al., 2021) and NDF (Sun et al., 2022a).

### S.3.2 ABLATION STUDY

We conduct an ablation study on the loss terms on the pediatric airway dataset. Airway shapes are more complicated than the *Starman* shapes and the hippocampi. Further, the number of shape samples is smallest among the three datasets. On this challenging dataset, our aim is to observe the model robustness when using varying loss terms and our goal is also to determine which loss terms are necessary and what suitable hyperparameter settings are.

Table S.12 and Table S.13 show the shape reconstruction evaluation for different hyperparameter settings. We see that $\mathcal{L}_{Dirichlet}$ for off-surface points is the most important term. A lower $\lambda_6$ for the latent code regularizer $\mathcal{L}_{lat}$ yields better reconstruction results. Table S.14 shows an ablation study for shape transfer. We observe that the reconstruction losses $\mathcal{L}_{Dirichlet}$ and $\mathcal{L}_{Neumann}$ are important for shape transfer.

To sum up, removing any of the reconstruction losses ($\mathcal{L}_{Eikonal}$, $\mathcal{L}_{Dirichlet}$, $\mathcal{L}_{Neumann}$) hurts performance. A smaller $\lambda_6$ yields better reconstruction performance, but may hurt shape transfer performance.

| Methods | variants | influenced term | CD ↓ | | EMD ↓ | | HD ↓ | |
|---|---|---|---|---|---|---|---|---|
| | | | $\mu$ | M | $\mu$ | M | $\mu$ | M |
| Ours (c) | $\lambda_1 = \lambda_5 = 0$ | $\mathcal{L}_{Eikonal}$ | 0.097 | 0.052 | 1.559 | 1.344 | 11.178 | 9.426 |
| Ours (c) | $\lambda_4 = 0$ | $\mathcal{L}_{Dirichlet\_off\_surf}$ | 4.029 | 4.485 | 1.454 | 1.239 | 68.272 | 69.356 |
| Ours (c) | $\lambda_3 = 0$ | $\mathcal{L}_{Neumann}$ | 0.089 | 0.055 | 1.494 | 1.313 | 10.52 | 8.625 |
| Ours (c) | $\lambda_2 = 0$ | $\mathcal{L}_{Dirichlet\_on\_surf}$ | 0.151 | 0.09 | 2.058 | 1.756 | 11.354 | 9.569 |
| Ours (c) | $\lambda_6 = 0$ | $\mathcal{L}_{lat}$ | 0.041 | 0.019 | 0.936 | 0.834 | 8.677 | 7.335 |
| Ours (c) | $\lambda_6* = 0.01$ | $\mathcal{L}_{lat}$ | 0.051 | 0.025 | 1.061 | 0.923 | 9.301 | 7.139 |
| Ours (c) | $\lambda_6* = 0.1$ | $\mathcal{L}_{lat}$ | 0.061 | 0.031 | 1.154 | 1.043 | 9.875 | 8.151 |
| Ours (c) | $\lambda_6* = 1$ | $\mathcal{L}_{lat}$ | 0.084 | 0.044 | 1.344 | 1.182 | 10.719 | 8.577 |
| Ours (c) | $\lambda_6* = 10$ | $\mathcal{L}_{lat}$ | 0.109 | 0.058 | 1.554 | 1.336 | 11.933 | 9.705 |
| Ours (c) | $\lambda_6* = 100$ | $\mathcal{L}_{lat}$ | 0.152 | 0.088 | 1.943 | 1.715 | 14.37 | 12.043 |

Table S.13: Ablation study: quantitative evaluation of shape reconstruction. Ours (c) means covariates are used as additional input to NAISR. $\mu$ indicates the mean value of the measurements; M indicates the median of the measurements. The shadowed line is what we report in the main text.

| Methods | ablations | | Volume Difference ↓ | | | |
| | variarants | influenced term | without covariates | | with covariates | |
| | | | $\mu$ | M | $\mu$ | M |
|---|---|---|---|---|---|---|
| Ours | $\lambda_1 = \lambda_5 = 0$ | $\mathcal{L}_{Eikonal}$ | 13.977 | 10.925 | 8.324 | 7.214 |
| Ours | $\lambda_4 = 0$ | $\mathcal{L}_{Dirichlet\_off\_surf}$ | 3736.527 | 3709.001 | 3693.514 | 3612.394 |
| Ours | $\lambda_3 = 0$ | $\mathcal{L}_{Neumann}$ | 24.717 | 24.877 | 26.094 | 25.311 |
| Ours | $\lambda_2 = 0$ | $\mathcal{L}_{Dirichlet\_on\_surf}$ | 55.579 | 57.766 | 64.845 | 63.442 |
| Ours | $\lambda_6 = 0$ | $\mathcal{L}_{lat}$ | 14.679 | 10.306 | 9.465 | 7.096 |
| Ours | $\lambda_6* = 0.01$ | $\mathcal{L}_{lat}$ | 8.766 | 6.629 | 8.518 | 5.105 |
| Ours | $\lambda_6* = 0.1$ | $\mathcal{L}_{lat}$ | 12.861 | 8.307 | 11.518 | 8.950 |
| Ours | $\lambda_6* = 1$ | $\mathcal{L}_{lat}$ | 12.820 | 8.837 | 11.227 | 9.653 |
| Ours | $\lambda_6* = 10$ | $\mathcal{L}_{lat}$ | 8.644 | 4.676 | 9.464 | 5.756 |
| Ours | $\lambda_6* = 100$ | $\mathcal{L}_{lat}$ | 11.959 | 8.857 | 11.939 | 8.113 |

Table S.14: Ablation study: quantitative evaluation of shape transfer. $\mu$ indicates the mean value of the measurements; M indicates the median of the measurements. The shadowed line is what we report in the main text.

| | $\mathcal{L}_{Dirichlet\_on\_surf}$ | $\mathcal{L}_{Dirichlet\_off\_surf}$ | $\mathcal{L}_{Neumann}$ | $\mathcal{L}_{Eikonal}$ | $\mathcal{L}_{lat}$ | CD↓ | | EMD↓ | | HD↓ | |
|---|---|---|---|---|---|---|---|---|---|---|---|
| | | | | | | $\mu$ | M | $\mu$ | M | $\mu$ | M |
| 0 | × 100 | × 0.01 | × 10 | × 10 | × 10 | 5.486 | 5.386 | 1.093 | 0.981 | 69.792 | 69.779 |
| 1 | × 0.1 | × 10 | × 1 | × 100 | × 0.01 | 2.046 | 1.657 | 9.576 | 8.835 | 33.779 | 30.757 |
| 2 | × 0.01 | × 100 | × 1 | × 0.1 | × 0.01 | 0.175 | 0.138 | 2.237 | 2.162 | 23.269 | 23.102 |
| 3 | × 0.1 | × 0.1 | × 0.01 | × 0.1 | × 100 | 0.168 | 0.131 | 1.697 | 1.641 | 15.570 | 14.062 |
| 4 | × 10 | × 0.01 | × 10 | × 0.01 | × 1 | 0.210 | 0.128 | 1.828 | 1.458 | 17.472 | 17.574 |
| 5 | × 10 | × 0.01 | × 0.1 | × 10 | × 10 | 3.815 | 4.483 | 1.538 | 1.437 | 61.785 | 68.420 |
| 6 | × 10 | × 0.01 | × 0.1 | × 0.1 | × 0.1 | 1.598 | 1.416 | 3.337 | 2.898 | 37.793 | 39.948 |
| 7 | × 0.01 | × 1 | × 100 | × 10 | × 10 | 33.090 | 25.892 | 0.028 | 0.027 | 86.968 | 80.081 |
| 8 | × 1 | × 100 | × 1 | × 0.01 | × 0.01 | 1.382 | 0.047 | 1.461 | 1.368 | 14.455 | 12.120 |
| 9 | × 100 | × 0.01 | × 100 | × 0.1 | × 100 | 0.202 | 0.103 | 1.837 | 1.453 | 20.760 | 19.042 |
| **10** | **× 0.1** | **× 1** | **× 1** | **× 0.01** | **× 0.1** | **0.066** | **0.038** | **1.309** | **1.150** | **10.494** | **8.781** |
| 11 | × 0.1 | × 0.1 | × 0.1 | × 10 | × 10 | 1.662 | 1.405 | 5.776 | 5.282 | 26.966 | 24.977 |
| 12 | × 1 | × 10 | × 0.01 | × 10 | × 100 | 0.193 | 0.126 | 2.662 | 2.419 | 13.704 | 11.538 |
| 13 | × 0.1 | × 1 | × 100 | × 10 | × 100 | 33.090 | 25.892 | 0.030 | 0.029 | 86.968 | 80.081 |
| **14** | **× 100** | **× 100** | **× 10** | **× 100** | **× 100** | **0.064** | **0.038** | **1.335** | **1.199** | **9.859** | **8.240** |
| 15 | × 100 | × 0.01 | × 100 | × 10 | × 1 | 4.302 | 4.304 | 1.216 | 1.020 | 63.564 | 62.475 |
| 16 | × 0.01 | × 0.1 | × 0.1 | × 10 | × 0.01 | 2.210 | 1.844 | 6.875 | 6.230 | 38.485 | 30.460 |
| 17 | × 0.01 | × 0.1 | × 1 | × 100 | × 1 | 5.560 | 5.916 | 6.561 | 6.098 | 62.960 | 67.416 |
| **18** | **× 0.01** | **×10** | **× 1** | **× 1** | **× 0.01** | **0.053** | **0.031** | **1.196** | **1.077** | **8.565** | **7.080** |
| 19 | × 0.1 | × 0.01 | × 1 | × 1 | × 10 | 0.510 | 0.166 | 2.905 | 2.099 | 25.228 | 15.788 |
| **Ours** | × 1 | × 1 | × 1 | × 1 | × 1 | **0.067** | **0.039** | **1.239** | **1.138** | **10.333** | **8.404** |

Table S.15: Ablation study: quantitative evaluation of shape reconstruction on random hyperparameter settings. $\mu$ indicates the mean value of the measurements; M indicates the median of the measurements. The shadowed line is what we report in the main text. Bold text indicates the hyperparameter settings which produce a relatively low chamfer distance (<0.1).

| | ablations | | | | | Volume Difference↓ | | | |
|---|---|---|---|---|---|---|---|---|---|
| Methods | $\mathcal{L}_{Dirichlet\_on\_surf}$ | $\mathcal{L}_{Dirichlet\_off\_surf}$ | $\mathcal{L}_{Neumann}$ | $\mathcal{L}_{Eikonal}$ | $\mathcal{L}_{lat}$ | without covariates | | with covariates | |
| | | | | | | $\mu$ | M | $\mu$ | M |
| 10 | × 0.1 | × 1 | × 1 | × 0.01 | × 0.1 | 10.891 | 8.929 | 11.800 | 9.095 |
| 14 | × 100 | × 100 | × 10 | × 100 | × 100 | 15.260 | 13.312 | 12.424 | 10.632 |
| 18 | × 0.01 | ×10 | × 1 | × 1 | × 0.01 | 18.164 | 17.261 | 20.980 | 19.932 |
| Ours | × 1 | × 1 | × 1 | × 1 | × 1 | 12.820 | 8.837 | 11.227 | 9.653 |

Table S.16: Ablation study: quantitative evaluation of shape transfer on successful randomized hyperparameter settings for reconstruction task. $\mu$ indicates the mean value of the measurements; M indicates the median of the measurements. The shadowed line is what we report in the main text.

Doing a full grid search over all 5 or 6 hyperparameters would be prohibitive. Instead, we test based on random configurations from the full hyperparameter grid. We tested 20 random configurations to 1) demonstrate our setting is reasonable; and to 2) provide some possible guidance from failure settings. Specifically, we created a grid by multiplying our 5 coefficients by $[0.01, 0.1, 1, 10, 100]$. Then, we randomly chose 20 grid points to test our model's robustness for different hyperparameter settings, as shown in Table S.15 and Table S.16. Our randomized grid-search analysis shows that our chosen hyperparameters provide good performance.

### S.3.3 SHAPE RECONSTRUCTION

Figure S.6 and Figure S.7 visualize more reconstructed hippocampi and airway shapes respectively. We observe that NAISR produces detailed and complete reconstructions from noisy and incomplete observations.

Each shape reconstruction method, except for A-SDF, successfully reconstructs airways and hippocampi. As discussed in Section 4.2, we suspect A-SDF overfits the training set by memorizing shapes with their covariates. We investigate this by evaluating shape reconstruction on the training set for A-SDF and NAISR as shown in Table S.17. From Table S.17, we can see that A-SDF overfits the training set.

### S.3.4 SHAPE TRANSFER

Table S.18 shows the transferred airways using NAISR without covariates as input (following Equation 10). The predicted shapes from Equation 9 and Equation 10 look consistent in terms of appearance and development tendency. Table S.19 and Table S.20 show the transferred hippocampi. Due to the limited observation time span of patients in the ADNI hippocampus dataset, the volume stays almost constant.

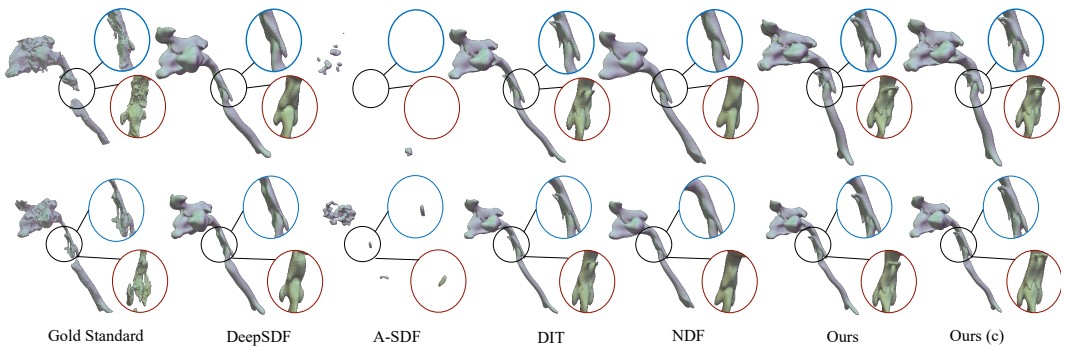

| Gold Standard | DeepSDF | A-SDF | DIT | NDF | Ours | Ours (c) |

Figure S.6: Visualizations of airway shape reconstructions with different methods. The red and blue circles show the structure in the black circle from two different views. **NAISR** produces detailed and accurate reconstructions and imputes missing airway parts.

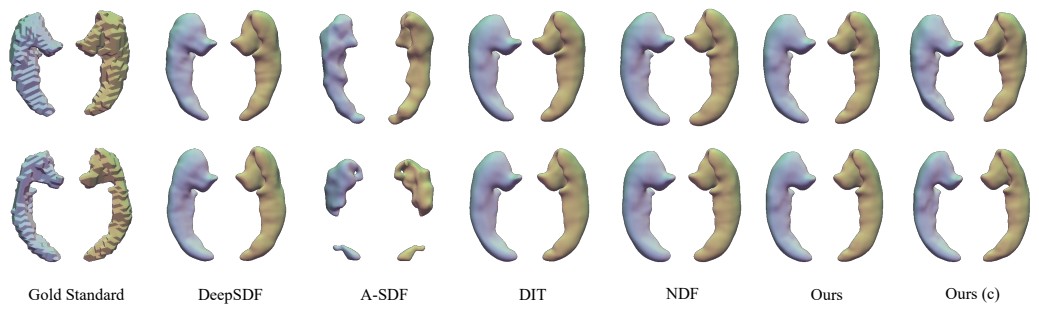

| Gold Standard | DeepSDF | A-SDF | DIT | NDF | Ours | Ours (c) |

Figure S.7: Visualizations of hippocampus shape reconstructions with different methods. The red and blue circles show the structure in the black circle from two different views. All methods except for A-SDF are able to reconstruct well.

### S.3.5 SHAPE DISENTANGLEMENT AND EVOLUTION

Fig. S.9 shows an example of airway shape extrapolation in covariate space for a patient in the testing set. Fig. S.10 shows an example of hippocampus shape extrapolation in covariate space for a patient in the testing set. We observe that in the range of observed covariates (inside the purple shade), shape extrapolation produces realistic-looking and reasonable growing/shrinking shapes in accordance with clinical expectations. Further, NAISR is able to extrapolate shapes outside this range, but the quality is lower than within the range of observed covariates.

### S.3.6 VISUALIZATION OF TEMPLATE LEARNING

Fig. S.11 shows the learned *Starman*, airway and hippocampus templates across epochs. We did not use a fixed atlas. Instead, the atlas is learned as the best shape explaining the shape population given the covariate-specific deformations. This estimated template shape is by-design expected to be the a population-average shape at the average age and average weight in the dataset. NAISR quickly converges and capture a reasonable template shape already within the first epochs.

| Methods | Training Set | | | | | | Testing Set | | | | | |
|---|---|---|---|---|---|---|---|---|---|---|---|---|
| | CD ↓ | | EMD ↓ | | HD ↓ | | CD ↓ | | EMD ↓ | | HD ↓ | |
| | $\mu$ | **M** | $\mu$ | **M** | $\mu$ | **M** | $\mu$ | **M** | $\mu$ | **M** | $\mu$ | **M** |
| A-SDF | 0.014 | 0.010 | 0.770 | 0.699 | 5.729 | 4.762 | 2.647 | 1.178 | 10.307 | 8.992 | 47.172 | 37.835 |
| Ours | 0.038 | 0.025 | 0.975 | 0.883 | 8.624 | 7.538 | 0.067 | 0.039 | 1.246 | 1.128 | 10.333 | 8.404 |

Table S.17: Comparison of NAISR and A-SDF on the training set. A-SDF performed well on the training set but failed on the testing set.

| #time | 0 | 1 | 2 | 3 | 4 | 5 | 6 | 7 | 8 | 9 | 10 |
|---|---|---|---|---|---|---|---|---|---|---|---|

| $\{\mathcal{S}^t\}$ | | | | | | | | | | | |
|---|---|---|---|---|---|---|---|---|---|---|---|
| # time | 0 | 1 | 2 | 3 | 4 | 5 | 6 | 7 | 8 | 9 | 10 |
| age | 154 | 155 | 157 | 159 | 163 | 164 | 167 | 170 | 194 | 227 | 233 |
| weight | 55.2 | 60.9 | 64.3 | 65.25 | 59.25 | 59.2 | 65.3 | 68 | 77.1 | 75.6 | 75.6 |
| sex | M | M | M | M | M | M | M | M | M | M | M |
| p-vol | 91.08 | 92.47 | 93.57 | 94.26 | 94.35 | 94.59 | 96.28 | 97.34 | 102.59 | 104.75 | 104.51 |
| m-vol | 86.33 | 82.66 | 63.23 | 90.65 | 98.11 | 84.35 | 94.14 | 127.45 | 98.81 | 100.17 | 113.84 |

Table S.18: Airway shape transfer without covariates as input for the patient shown in the main text. Blue: gold standard shapes; red: transferred shapes with NAISR. The table below lists the covariates (age/month, weight/kg, sex) for the shapes above. P-vol(predicted volume) is the volume ($cm^3$) of the transferred shape by NAISR with covariates following Eq. equation 9. M-vol (measured volume) is the volume ($cm^3$) of the shapes based on the actual imaging. The transferred shapes show similar growth trends in pediatric airways as shown in Table. 4.

| # time | 0 | 1 | 2 | 3 | 4 |
|---|---|---|---|---|---|

| $\{\mathcal{S}^t\}$ | | | | | |
|---|---|---|---|---|---|
| # time | 0 | 1 | 2 | 3 | 4 |
| age | 64.7 | 65.2 | 65.2 | 65.7 | 65.7 |
| AD | No | No | No | No | No |
| sex | F | F | F | F | F |
| edu | 14 | 14 | 14 | 14 | 14 |
| p-vol | 1.55 | 1.55 | 1.55 | 1.55 | 1.55 |
| m-vol | 1.49 | 1.56 | 1.55 | 1.45 | 1.55 |

Table S.19: Hippocampus shape transfer with covariates as input. Blue: gold standard shapes; red: transferred shapes with NAISR. The table below lists the covariates (age/yrs, AD, sex, edu(education length)/yrs for the shapes above. P-vol(predicted volume) is the volume ($cm^3$) of the transferred shape by NAISR with covariates following Equation 10. M-vol (measured volume) is the volume ($cm^3$) of the shapes based on the actual imaging. The transferred shapes stay almost the same in the one-year time period for this patient.

| # time | 0 | 1 | 2 | 3 | 4 |
|--------|---|---|---|---|---|

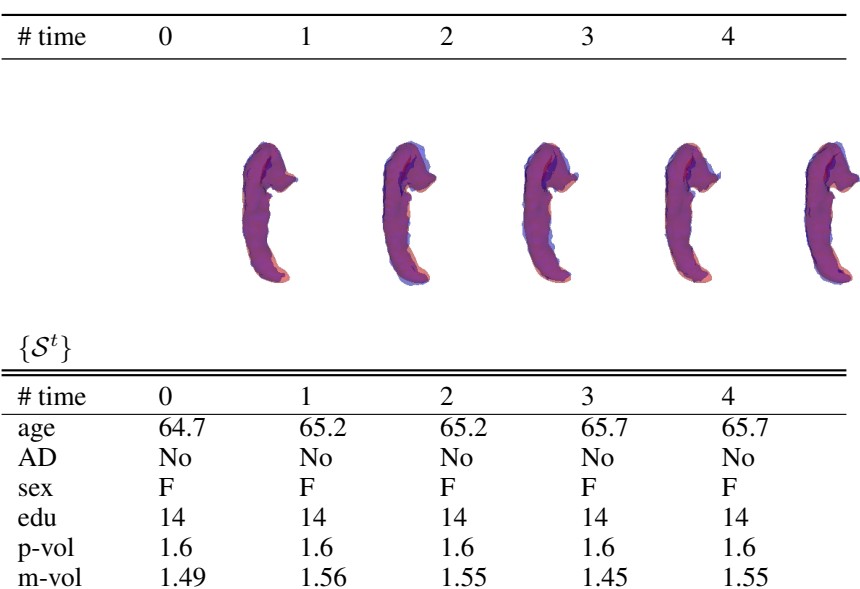

$\{\mathcal{S}^t\}$

| # time | 0 | 1 | 2 | 3 | 4 |
|--------|------|------|------|------|------|
| age | 64.7 | 65.2 | 65.2 | 65.7 | 65.7 |
| AD | No | No | No | No | No |
| sex | F | F | F | F | F |
| edu | 14 | 14 | 14 | 14 | 14 |
| p-vol | 1.6 | 1.6 | 1.6 | 1.6 | 1.6 |
| m-vol | 1.49 | 1.56 | 1.55 | 1.45 | 1.55 |

Table S.20: Hippocampus shape transfer without covariates as input. Blue: gold standard shapes; red: transferred shapes with NAISR. The table below lists the covariates (age/yrs, AD, sex, edu(education length)/yrs for the shapes above. P-vol(predicted volume) is the volume ($cm^3$) of the transferred shape by NAISR with covariates following Equation 9. M-vol (measured volume) is the volume ($cm^3$) of the shapes based on the actual imaging. The transferred shapes stay almost the same in the one-year period space for this patient.

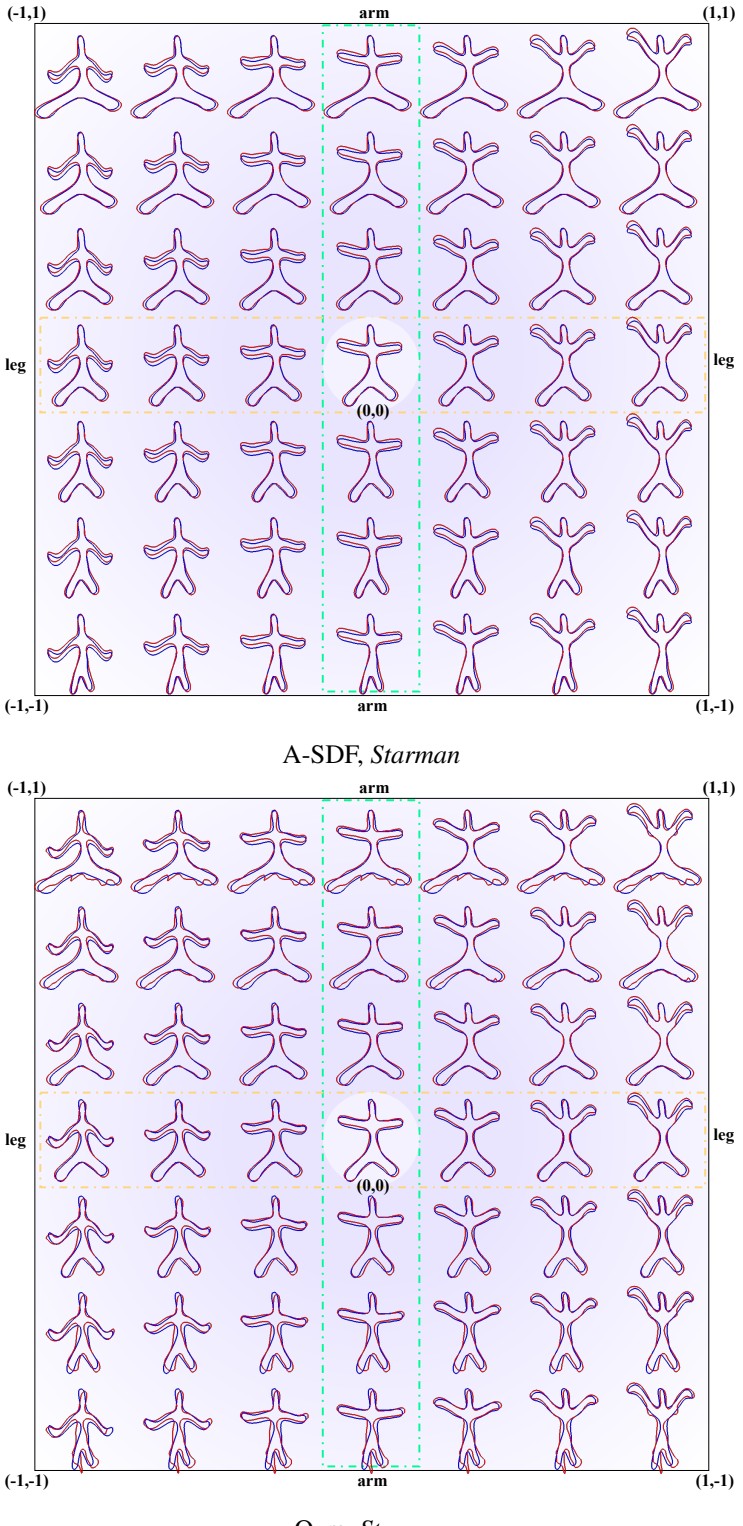

Figure S.8: Individualized *Starman* shape extrapolation in covariate space. The blue shapes are the groundtruth shapes and the red shapes are the reconstructions. The purple shadows over the space indicate the covariate range that the dataset covers. The latent code **z** is kept constant to create an individualized covariate shape space. The shapes in the green and yellow boxes are plotted with $\{\Phi_i\}$ (see Section 3.4), representing the disentangled shape evolutions along the arm and leg respectively. Shapes extrapolated from $\mathbf{z}_i$ look realistic and smooth across different covariates.

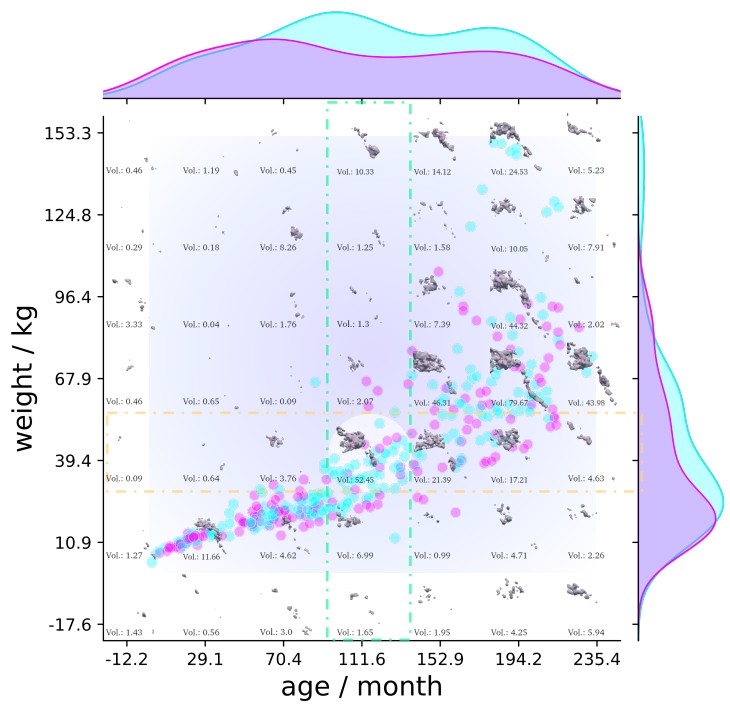

A-SDF, Pediatric Airway

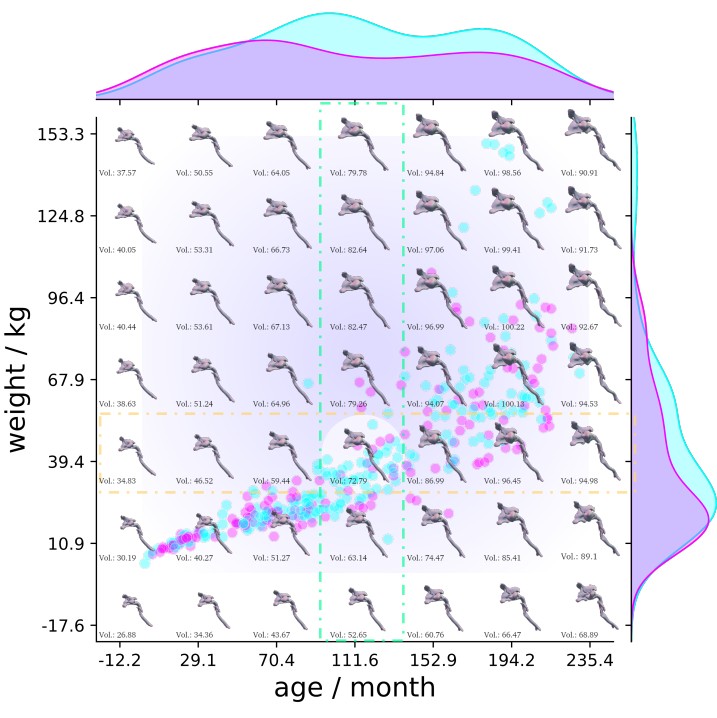

Ours, Pediatric Airway

Figure S.9: Individualized airway shape extrapolation in covariate space. Example shapes in the covariate space are visualized with their volumes ($cm^3$) below. Cyan points represent male and purple points female children in the dataset. The points represent the covariates of all children in the dataset. The purple shadows over the space indicate the covariate range that the dataset covers. The colored shades at the boundary represent the covariate distributions stratified by sex. The latent code $\mathbf{z}$ is kept constant to create an individualized covariate shape space. The shapes in the green and yellow boxes are plotted with $\{\Phi_i\}$ (see Section 3.4), representing the disentangled shape evolutions along weight and age respectively. Shapes extrapolated from $\mathbf{z}_i$ look realistic and smooth across different covariates.

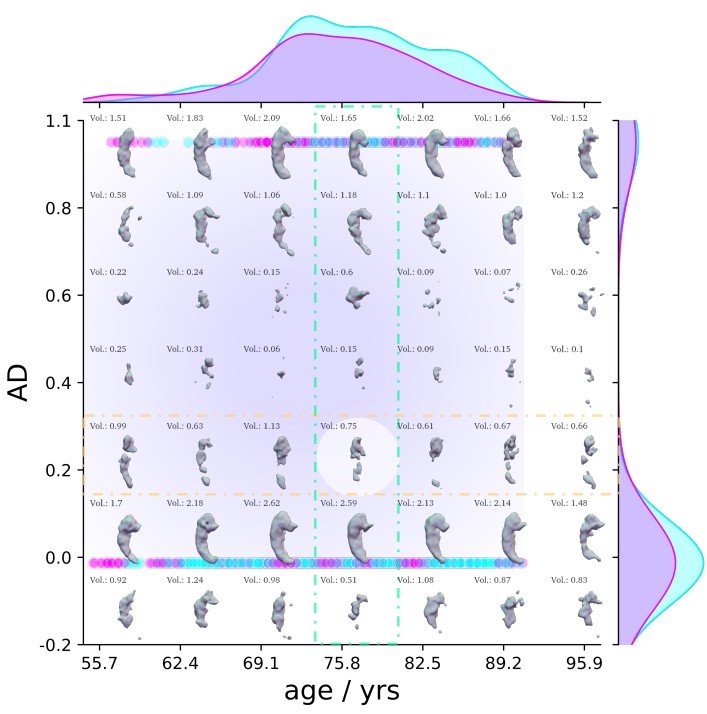

A-SDF, ADNI Hippocampus

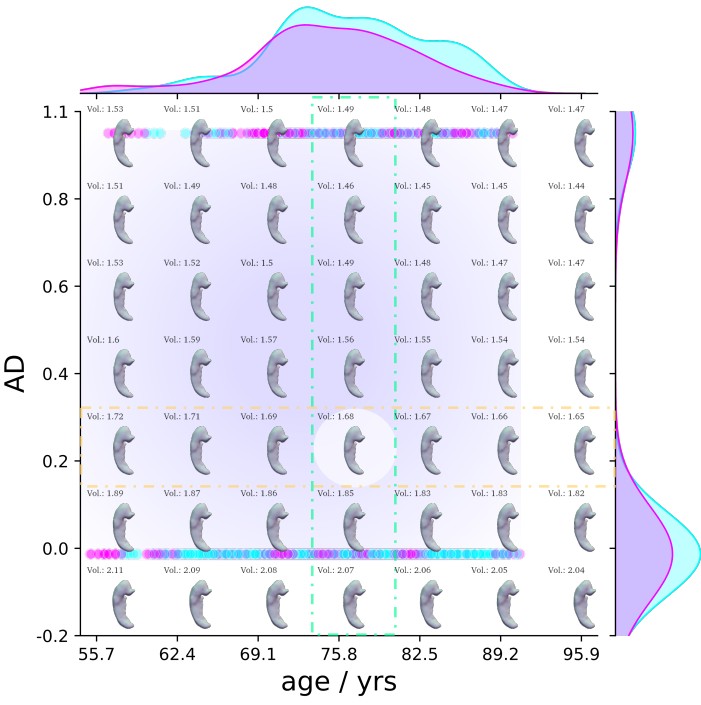

Ours, ADNI Hippocampus

Figure S.10: Individualized hippocampus shape extrapolation in covariate space. Example shapes in the covariate space are visualized with their volumes ($cm^3$) below. Cyan points represent male and purple points female patients in the dataset. The points represent the covariates of all patients in the dataset. The purple shadows over the space indicate the covariate range that the dataset covers. The colored shades at the boundary represent the covariate distributions stratified by sex. The latent code $\mathbf{z}$ is kept constant to create an individualized covariate shape space. The shapes in the green and yellow boxes are plotted with $\{\Phi_i\}$ (see Section 3.4), representing the disentangled shape evolutions along AD and age respectively. Shapes extrapolated from $\mathbf{z}_i$ look realistic and smooth across different covariates.

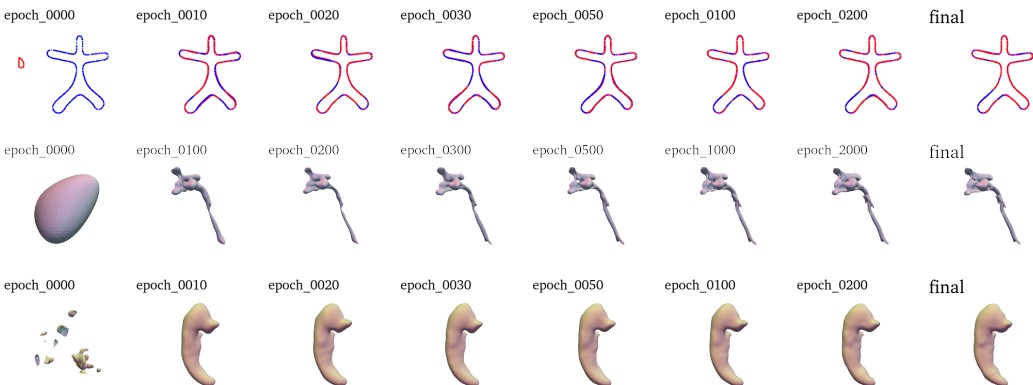

Figure S.11: Learned *Starman*, airway, and hippocampus templates across epochs. The blue *Starman*s are the ground truth while the red ones are our learned templates across epochs.

