# OpenReview forum: "$\texttt{NAISR}$: A 3D Neural Additive Model for Interpretable Shape Representation"
_ICLR.cc/2024/Conference — ICLR 2024 spotlight_

### Official Review · Reviewer_ftBT · 2023-10-29

**Soundness:** 3 good
**Presentation:** 4 excellent
**Contribution:** 3 good
**Rating:** 8
**Confidence:** 5

**Summary:**

The paper introduces NAISR, a 3D neural additive model that aims to provide an interpretable shape representation. This approach combines deep implicit shape representations with an atlas that deforms in response to specified covariates. The utility of NAISR is demonstrated through evaluations on simulated and real medical datasets. The paper discusses the advantages of NAISR over other shape representation methods, particularly in capturing individual covariates' effects on shapes, shape transfer capabilities, and the ability to generate shapes based on extrapolated covariates.

**Strengths:**

- The paper is well-structured, making it easy to follow. The flow of content is logical, facilitating a clear understanding of the proposed model and its implications.

- The paper tackles a significant gap in the shape modeling field: quantitatively capturing shape changes concerning problem-specific covariates. This is a significant contribution to the domain of shape representation and analysis.

- Leveraging neural additive deep implicit for shape representation offers interpretability and shape reconstructions that are resolution agnostic.

- The proposed approach's efficacy is showcased on both simulated data and medical datasets.

- The paper compares NAISR with existing state-of-the-art implicit shape representation methods, further establishing its superior performance and potential for broader applications.

**Weaknesses:**

- The paper assumes the given population to be unimodal and representable fully by a learned atlas. However, the methodology's ability to handle multimodal distributions and topological variations remains unclear.

- The experiments do not adequately demonstrate the model's performance under limited training sample sizes, a typical scenario in medical shape analysis.

- While deformations from the learned atlas provide a means to statistical shape analysis, the paper does not evaluate or showcase the learned models' statistical aspects. This omission limits the paper's depth and breadth in understanding the model's holistic performance.

- The paper does not discuss the sensitivity of the proposed model to hyperparameters, which could be crucial for replication and application in various contexts.

Minor: Authors provide a discussion of limitations in the supplementary material, however this should be part of the main paper.

**Questions:**

- It is unclear how the atlas is initialized. It would be very helpful to show the learned atlas across epochs.

- The paper does not discuss the computational complexity and runtime for various processes, including training, inference, shape transfer, evolution, and disentanglement. Given that neural implicit representation works at the point coordinate level, understanding these metrics is crucial.

-  What are the computational and time requirements to reconstruct a full shape using the proposed implicit representation?

- The paper does not adequately justify the use of a non-amortized latent code, particularly the latent code encapsulating shape parameters not observed or captured by the covariate. The latent code is being optimized during training and inference as opposed to using a neural network to estimate such latent codes.

---

> ### Author Response · Authors · 2023-11-17
> **Answer to Reviewer 4 - Part 1**
>
> #### **The paper assumes the given population to be unimodal and representable fully by a learned atlas. However, the methodology's ability to handle multimodal distributions and topological variations remains unclear.**
>
> The current approach is not targeted at problems with changing topology as it is based on deforming an estimated template image. We will clarify this in the paper. This property might be a restriction in certain application contexts and could, for example, be overcome by allowing subject-specific modifications to the template level set function. However, note that in medical applications preserving topology is often desirable. Such applications are currently our main target. Our model currently also only considers one template shape. However, extensions to multiple templates, which would then be selected based on the best fit for a specific subject, are conceivable. Note though that while supporting multiple templates would be a more flexible approach for shape fitting it would at the same time complicate the construction of spatial correspondences between all shapes which is often a primary requirement in medical shape analysis. We will add related discussions to the paper.
>
>
>
> #### **The experiments do not adequately demonstrate the model's performance under limited training sample sizes, a typical scenario in medical shape analysis.**
>
> We conducted an experiment to train $\texttt{NAISR}$ with only 50 airway shapes and found $\texttt{NAISR}$ fails to work on such a limited sample size with a median Chanfer distance of 0.7440 (0.0390 using the full training set).
>
> Very limited sample sizes are likely not a good match for our approach as our goal is to not only capture a shape space for a population but also to capture variability with respect to given covariates. Hence, a sufficient number of observations across the desired covariate range is desirable. The smallest dataset we used (the pediatric airway dataset) contains 357 shapes, which is a medium-sized dataset. However, we argue that this dataset is challenging and representative for medical scenarios because 1) most subjects (n=229) only have one time observation; and 2) large shape and size variations can be observed as the dataset spans a large age range from infant to adult.
>
> Our experiments show that the required sample sizes can be much smaller than the methods designed for general computer vision datasets, such as A-SDF, which we believe is a significant improvement. However, they cannot be too small based on the results using only 50 samples discussed above.
>
>
> #### **While deformations from the learned atlas provide a means to statistical shape analysis, the paper does not evaluate or showcase the learned models' statistical aspects. This omission limits the paper's depth and breadth in understanding the model's holistic performance.**
>
> We are not sure we are following this comment. Our present goal was to reliably capture shape changes related to the provided covariates. Future work will focus on using such relationships for example to score disease severity with respect to a normal captured population. For example, for the airway dataset this would amount to comparing the normal airway for a particular age, weight, and sex (as captured by $\texttt{NAISR}$) to the airway of a specific child at that age to understand how the airway would differ. Including such analyses in the present paper would likely distract from the current focus on developing the base $\texttt{NAISR}$ model. We would be happy to add any general analyses that would help assess the statistical aspects. What are your specific recommendations? What would you like to see?
>
> #### **The paper does not discuss the sensitivity of the proposed model to hyperparameters, which could be crucial for replication and application in various contexts.**
>
> We thank the reviewer for pointing this good question out.
>
> The ablation study in Tables S.12 and S.13 shows that removing any of the reconstruction losses ($\mathcal{L}\_{Eikonal}$, $\mathcal{L}\_{Dirichlet}$, $\mathcal{L}\_{Neumann}$) hurts performance.
>
> We did not perform a full grid-search to find the best hyperparameters due to the size of this space. However, we now verified based on a random grid research that our chosen  model parameters give good performance. See Table S.15 and Table S.16 in the supplementary material. See also the related question by reviewer 2.
>
> Our observation is that the reconstruction terms $\mathcal{L}\_{Dirichlet}$ (the penalty for signed distances to the surfaces) and $\mathcal{L}\_{Neumann}$ (the penalty for consistency with normal vectors) should be kept high enough to produce good reconstructions.  Values for $\lambda_6$ (for regularizing latent codes) that are too small will decrease the shape transfer performance.

---

> > ### Comment · Reviewer_7YZF · 2023-11-18
> >
> > I appreciate the detailed feedback that addressed most of my concerns. I agree with other reviewers that it would be very beneficial if you could succinctly state the limitations of the method and also place them into the main manuscript.

---

> ### Author Response · Authors · 2023-11-18
> **Answer to Reviewer 4 - Part 2**
>
> #### **Minor: Authors provide a discussion of limitations in the supplementary material, however, this should be part of the main paper.**
>
> Thanks for the advice. We will include a short discussion on limitations (due to page constraints) in the main manuscript.
>
> #### **It is unclear how the atlas is initialized. It would be very helpful to show the learned atlas across epochs.**
>
> Thanks you for the suggestion. We have added Figure S.11 to the supplementary material, showing the atlas (template shape) learning progress. This figure shows that the atlas shape quickly converges. The atlas network and the deformation network both follow the SIREN initialization, as illustrated in S.3.1 in the supplementary material. We did not start with a fixed atlas. Instead, the atlas is purely learned as the average shape from the shape population, which is by design expected to be the average shape at the average age and average weight in the dataset.
>
> #### **The paper does not discuss the computational complexity and runtime for various processes, including training, inference, shape transfer, evolution, and disentanglement. Given that neural implicit representation works at the point coordinate level, understanding these metrics is crucial.**
>
> Thank you for the suggestion. We now measured runtimes and have added these measures to our paper in S.3.1 (see the paragraph on computational runtime) in the supplementary material.
>
> #### **What are the computational and time requirements to reconstruct a full shape using the proposed implicit representation?**
>
> The model is trained on an Nvidia GeForce RTX 3090 GPU approximately for 12 hours for 3000 epochs for the pediatric airway dataset, and for around 3.5 hours for 300 epochs for the ADNI hippocampus dataset. It takes around 130 seconds for 800 iterations and 30 seconds for 200 iterations to infer the latent code $\mathbf{z}$ and the covariates $\mathbf{c}$ respectively. However, we found 200 iterations should be sufficient to produce a reasonable reconstruction in most cases. Then, it takes approximately 30 seconds to sample the SDF field and apply Marching Cubes to extract the shape mesh. Regarding shape transfer, evolution, and disentanglement for a specific case, we first need to optimize the latent code $\mathbf{z}$ as in shape reconstruction. Once the latent code $\mathbf{z}$ is optimized or assigned (e.g., to $\mathbf{0}$ as a template), it will take around 30 seconds to produce a new mesh controlled by the covariate $\mathbf{c}$.
>
> #### **The paper does not adequately justify the use of a non-amortized latent code, particularly the latent code encapsulating shape parameters not observed or captured by the covariate. The latent code is being optimized during training and inference as opposed to using a neural network to estimate such latent codes.**
>
> Our method derives from DeepSDF, the pioneering approach in using a coordinate-based neural network to represent shapes, with an auto-decoder to infer the latent code $\mathbf{z}$ during the testing stage. The reason for using an auto-decoder instead of an encoder is that it is difficult to make sure the high-dimensional latent space produced by the encoder is condensed yet sufficient to represent realistic unseen shapes (as discussed in the supplementary material of the DeepSDF paper). A powerful encoder requires the model to have generative ability. However, in our case, we may only obtain several hundreds of airway shapes in the medical dataset, which makes it challenging to obtain a powerful encoder. Therefore, we choose to perform test time inference to obtain $\mathbf{z}$, which can prevent us from the risk of overfitting an auto-encoder while still retaining all of our desired capabilities of the model.

---

> ### Author Response · Authors · 2023-11-18
> **Answer to Reviewer 7YZF - Thank you**
>
> You are welcome. We thank you for your timely feedback and good suggestions! We will add a succinct limitation section in the main manuscript. Space is tight, but we will make it fit.

---

> ### Author Response · Authors · 2023-11-22
> **Kind Reminder**
>
> Dear Reviewer ftBT,
>
> $~$
>
> Thank you again for your valuable feedback and comments! As the discussion period is approaching its end in 18 hours, we would greatly appreciate it if you could let us know whether our rebuttal and the updated manuscript addressed your concerns and if so, if it's possible to increase your rating. We are happy to address any of your remaining concerns.
>
> $~$
>
> Sincerely,
>
> Authors of Paper 2838

---

> > ### Comment · Reviewer_ftBT · 2023-11-22
> >
> > Thanks to the authors for their detailed and insightful response to the concerns I raised. After thoroughly considering your explanations and reflecting on the input from other reviews, I have decided to increase my original score.

---

> > > ### Author Response · Authors · 2023-11-23
> > > **Thank you**
> > >
> > > Thanks for your timely feedback and for raising the score. We appreciate your time and patience in reviewing our paper!

---

### Official Review · Reviewer_7YZF · 2023-10-30

**Soundness:** 4 excellent
**Presentation:** 4 excellent
**Contribution:** 2 fair
**Rating:** 6
**Confidence:** 4

**Summary:**

The paper proposes a concept disentangled deep implicit representation learning method for 3D shapes, with a focus on medical imaging application. Given a set of shapes from the same class represented by a point cloud, the proposed method identifies a template, and learns a set of displacement maps with respect to both known and unknown covariates.

The technique is compared to five other implicit shape representations (based on implicit functions), and qualitatively and quantitatively outperforms the best competitor A-SDF (Mu 2021), which also learns disentangled implicit shape representation.

Experiments are conducted on Starman, a simulated 2D shape dataset (5041 training + 4966 testing shapes), ADNI hippocampus 3D shape dataset (1632 hippocampus shapes segmented from magnetic resonance images (MRI)), and pediatric airway 3D shape dataset (357 airway shapes segmented from computed tomography (CT)). Qualitative and quantitative results conducted using shape reconstruction, transfer, disentanglement and evolution experiments and demonstrate competitive performance.

EDIT: I have updated my rating after reading authors' comments.

**Strengths:**

The paper is well-written and easy to follow. Experiments are thorough and conducted with well-established metrics (Chamfer distance (CD), Earth mover’s distance (EMD), Hausdorff distance (HD)), all established metrics for evaluating 3D shape methods. Code is provided in supplementary materials.

Visualizations are very clear, and lots of additional details are provided in supplementary material.

**Weaknesses:**

It is not clear how SD based techniques will be beneficial for medical images. Majority of medical imaging techniques use voxel-based representations. The proposed implicit shape representation (as other baselines) smoothed out details (compare ‘’gold standard’’ to all methods in Figure 2, for instance) and it not clear why a method with such artifacts would be beneficial in practice? Authors should provide more clear motivation of why the proposed method will be useful for medical imaging.

As described in Section 3.1, it is assumed that “the overall displacement field is the sum of displacement fields that are controlled by individual parameters”, plus there is a contribution of the unknown covariate (z) which cannot be controlled. However, it isn’t clear what happens if a key covariate is unknown (e.g., not provided in metadata) or a not useful covariate is present? The authors should clearly explain implications of such assumptions.

There are several weaknesses in related work:

* Relationship to other methods (Mu 2021) is not clearly described. What is the methodological advantage that the proposed technique brings?
* Related work is limited to representations focusing on implicit functions. There is other work in allowing parametrized shape editing, e.g., see for instance: Learning to Infer Semantic Parameters for 3D Shape Editing (Wei, 3DV 2020)

**Questions:**

In section 4.2, authors describe experiments with shape reconstruction, and indicate that the the proposed method and other SDF techniques can help complete shapes. It isn’t clear whether the technique is trained for shape completion, and if so, where do the partially complete shape come from? Does the fact that a shape is partial negatively influence method training?

**Details Of Ethics Concerns:**

No ethics concerns.

---

> ### Author Response · Authors · 2023-11-17
> **Answer to Reviewer 3 - Part 1**
>
> We thank the reviewer for the time, advice, and feedback. We address the main questions below:
>
> #### **It is not clear how SD based techniques will be beneficial for medical images. Majority of medical imaging techniques use voxel-based representations. The proposed implicit shape representation (as other baselines) smoothed out details (compare ''gold standard'' to all methods in Figure 2, for instance) and it not clear why a method with such artifacts would be beneficial in practice? Authors should provide more clear motivation of why the proposed method will be useful for medical imaging.**
>
> This is a good question. We are excited to share the background and motivation of our research problem which motivates $\texttt{NAISR}$.
>
> * While medical images are volumetric, shape analysis based on shapes extracted from volumetric segmentations is an important task in medical image analysis to understand shape differences. These shapes are, of course, derived from the volumetric images. In our case, the objective is to capture normal airway shape development, w.r.t, age, and weight, as little is known about it, and understanding it better will be able to assess airway abnormalities. Think about it as the equivalent of a growth chart for children. Here we simply do not limit ourselves to height but want to capture shape change with age.
>
> * In many shape analysis problems, acounting for covariates is important, because the shape population contains large variations and the covariates help with indicating which part of the shape population we should look at when analyzing new cases. Take our pediatric airway shape dataset as an example: if we want to characterize airway abnormalities for a 10-year-old child, airways of newborn babies and adults will be quite different. In our case, we use $\texttt{NAISR}$  to learn how the airway shape changes based on the covariates (e.g., age and weight).
>
> We agree that there is a smoothing effect. However, we believe that this smoothing effect can be beneficial (and should therefore not be considered an artifact) based on two reasons:
>
> * The jagged appearance of the gold standard shapes in Figure 2 is due to the voxel representation of the underlying segmentations. Hence, these are not representations of the true geometry as, for example, real hippocampi or airways are smooth. In fact, in classical medical shape analysis, shapes are commonly smoothed before analyses or represented by smooth basis function (for example, based on spherical harmonics). Our approach naturally provides a reasonable level of smoothing preserving key details (as can be seen in Figure 2) while reducing voxelization effects. Note also that in real clinical data voxel resolutions may differ between image acquisitions: this is for example the case for the airway shapes. Hence, removing these voxelization artifacts is desirable. It is true that strong geometric shape distortions might be problematic, but this is not what we observe. Ultimately, the success of these shape approximations should be judged in the context of a downstream task as part of follow-up work.
>
> * The core question we are trying to answer with $\texttt{NAISR}$ is, how an average shape (atlas shape) for particular covariates looks like. This population average is expected to be smooth.

---

> ### Author Response · Authors · 2023-11-17
> **Answer to Reviewer 3 - Part 2**
>
> #### **As described in Section 3.1, it is assumed that "the overall displacement field is the sum of displacement fields that are controlled by individual parameters", plus there is a contribution of the unknown covariate (z) which cannot be controlled. However, it isn't clear what happens if a key covariate is unknown (e.g., not provided in metadata) or a not useful covariate is present? The authors should clearly explain implications of such assumptions.**
>
> We thank the reviewer for posing this excellent question. This may happen in common clinical scenarios. There are two scenarios to be discussed here. **The first scenario is that the covariate is known at the training stage but missing at the test stage.** We discussed this in Sec.3.4, mainly in the Shape Reconstruction and Shape transfer part, but provide more details below which we will use to improve the discussion in the paper.
>
> Suppose we already have a trained $\texttt{NAISR}$ model $\Phi$ and $\Phi$ has been trained well to represent population-level shape variations with respect to the covariates.
>
> For a new shape $s_{tgt}$ with unknown covariates, we first need to recover its corresponding latent code $\mathbf{z}$ and the covariates $\mathbf{c}$. To estimate these quantities, the network parameters stay fixed and we optimize over the covariates $\mathbf{c}$ and the latent code $\mathbf{z}$ which are both randomly initialized. Specifically, we solve the optimization problem,
>
> $$\hat{\mathbf{c}},\hat{\mathbf{z}}=\underset{\mathbf{c},\mathbf{z}}{\arg \min }~\mathcal{L}(\Phi, \mathbf{c}, \mathbf{z})$$
>
> If the covariates  (e.g., age or weight at imaging time) are known we only infer the latent code $\mathbf{z}$ by optimizing
>
> $$\hat{\mathbf{z}}=\underset{\mathbf{z}}{\arg \min }~\mathcal{L}(\Phi,\mathbf{c}, \mathbf{z})\$$
>
> A new patient shape with different covariates can then be generated by extracting the zero level set of $\Phi(\mathbf{p}, \mathbf{c}_{\mathrm{new}}, \hat{\mathbf{z}})$.
>
> **The second scenario is where a key covariate is never recognized and thus not modeled as a covariate with $\texttt{NAISR}$.** In this case the deformation will purely be driven by the latent code $\mathbf{z}$. If we never capture an important covariate, $\texttt{NAISR}$ will gracefully  revert to a simple deformation-based shape representation model, from which we can still obtain shape reconstructions but which cannot be used to observe the missing covariates' effects of shapes.

---

> > ### Author Response · Authors · 2023-11-17
> > **Answer to Reviewer 3 - Part 3**
> >
> > #### **Relationship to other methods (Mu 2021) is not clearly described. What is the methodological advantage that the proposed technique brings? Related work is limited to representations focusing on implicit functions. There is other work in allowing parametrized shape editing, e.g., see for instance: Learning to Infer Semantic Parameters for 3D Shape Editing (Wei, 3DV 2020)**
> >
> >
> > We thank the reviewer for this question. We will update our manuscript to better clarify the motivation of our method. We apologize for not being able to include the full version of our research survey in the main text due to the 9-page limit. A more complete version of the research survey is provided in supplementary material S.1., including discussions of **deep implicit functions**, **point correspondences**, **disentangled representation learning**, **articulated shapes**, and **explainable artificial intelligence**, as well as how these methods relate to yet differ from our method.
> >
> > Regarding the methodological advantage of $\texttt{NAISR}$ over other methods, we provide comparisons in Table 1 in the main text and discuss our experimental observations in Section 4.2 and in the supplementary material (S.3.3). Among all the comparison methods, only A-SDF (Mu 2021) is suitable for similar tasks (shape transfer and disentangled evolution) as our method.
> >
> > However, A-SDF 1) does not consider deformations; 2) uses covariates as different inputs into one MLP which are therefore entangled together inside the network; 3) assumes that each articulation affects a separate object part while in medical scenarios, covariates often affect shapes in a more entangled and complex way; 4) is trained on a large longitudinal dataset, where each shape has at least 31 different articulations (e.g., a single door has 31 observations for different door opening angles.) On the other hand, $\texttt{NAISR}$ is a deformable shape representation that can work on smaller shape datasets and disentangle complex covariate interactions based on the neural additive structure of the deformation model.
> >
> > The goal of the work by Wei et al. is semantic editing of geometric shapes. For example one might have an input chair represented as a point cloud for which we want to adjust the height of the back. To be able to do such semantic editing operations requires a set of defined shape attributes that correspond to our covariates. An encoder is then trained to predict shape attributes from a given point cloud. The decoder to go from an attribute vector to a shape is not learned however, but instead the geometric deformation model is assumed known and applied to a known template shape, termed an analytical decoder. Hence, the approach easily allows synthesizing synthetic shapes for different attribute vectors. As it is assumed known how a given shape is deformed by a set of attributes it is possible to create deformed shapes for two different attribute vectors and then to directly infer a deformation field between the two shapes (as both shapes come from the same template and we therefore know the point-to-point correspondences).
> >
> > Transferring such an attribute change to a real, non-synthetic input point cloud then corresponds to predicting the attributes for this shape. Two synthetic shapes and their corresponding deformation are then created based on the predicted attribute vector and the edited predicted attribute vector (for example, by adjusting the attribute for chair back height). The corresponding deformation is then applied to the input shape. While our approach has conceptual similarities it is ultimately significantly different in the following ways: 1) it estimates a template shape and does not assume one; 2) it estimates the deformation models, i.e., how the covariates deform the template shape, rather than assuming it is known how the covariates affect the deformations; and 3) it is not based on simulated deformations.
> >
> > The problem settings for the works by Mu et al. and Wei et al. do not carry over to typical scientific shape discovery scenarios because 1) they in general do not provide densely sampled longitudinal data for every patient; and 2) sample sizes are usually relatively small. For example, in our airway dataset, most kids only have one observation (Table.S.7). Our experiments show that A-SDF (Mu 2021) works exceptionally well at reconstructing shapes in the training set but does not work well for unseen airways in the testing set, indicating that A-SDF (Mu 2021) overfits the training set by memorizing shapes with their covariates. In contrast, $\texttt{NAISR}$ is designed for realistic scientific shape discovery problems and works for small and medium-sized datasets.

---

> > > ### Author Response · Authors · 2023-11-17
> > > **Answer to Reviewer 3 - Part 4**
> > >
> > > #### **In section 4.2, authors describe experiments with shape reconstruction, and indicate that the the proposed method and other SDF techniques can help complete shapes. It isn't clear whether the technique is trained for shape completion, and if so, where do the partially complete shape come from? Does the fact that a shape is partial negatively influence method training?**
> > >
> > >
> > > As proposed in the DeepSDF paper, one advantage of implicit shape representations is that inference can be performed from an arbitrary number of SDF samples. Shape completion under such implicit frameworks amounts to solving for the shape code $\mathbf{z}$ that best explains a partial shape observation via Eq. (9) or Eq. (10). Given the shape code a complete shape can be rendered as the decoder implicitly captures the full shapes.
> > >
> > > In our pediatric airway dataset, partial shapes exist during data collection due to differing fields of views during image acquisition. For example, some images contain the carina (the first branch of the airway) and others do not. Analyzing partial shapes is therefore inevitable in our research problem.
> > >
> > > Our method is not trained specifically for shape completion. Instead, shape completion is an additional benefit we obtain from the implicit shape representation design. Different from voxel-based representations, an individual point is the input to the network, instead of a volume, meaning that it can be evaluated anywhere and extrapolation (within reason) is possible.

---

> > > ### Comment · Reviewer_7YZF · 2023-11-18
> > > **Clarification on sample size**
> > >
> > > Thank you for the detailed response. Could you please clarify what the distinction of the proposed method that allows it not to overfit or to better tackle smaller sets, in comparison to prior work? I am confused due to the comment to R4 that "sample sizes are likely not a good match for our approach".

---

> ### Author Response · Authors · 2023-11-18
> **Answer to Reviewer 3 - for clarification on sample size**
>
> Thank you for your excellent question. Sorry for any confusion we created with the answer to R4 that “very limited sample sizes are likely not a good match for our approach.” What we meant is that if sample sizes get much too small our approach will, of course, also start struggling to fit a model (other models would too), because we are attempting to model possibly complex dependencies of the covariates on shapes and therefore need to rely on a reasonably dense sampling of the covariate space. What should be considered reasonably dense is likely application-dependent (and would be good to explore going forward), but if the sampling gets very sparse the model would for example need to rely on extrapolations outside the observed covariate range. Note that the most comparable model to ours is A-SDF and our experiments show that NAISR results in better shape modeling capabilities for the sample sizes we investigate (e.g., n=357 for the airway data). Hence, we empirically observed that our model works better than others for small datasets.
>
> We believe the reason for the better behavior of NAISR than other models for smaller sample sizes is based on our additive deformation design, which distinguishes it for example from A-SDF. Suppose we only have limited observations of shapes and each is associated with multiple covariates $\mathbf{c}=\lbrace{c_{i, i=1,...,n}}\rbrace$. Existing neural shape modeling approaches which may apply to our problems use these covariates $\mathbf{c}=\lbrace{c_{i, i=1,...,n}}\rbrace$ as different inputs into one neural network (e.g., MLP). This then puts the burden on the neural network to determine how these dependencies affect shape. Our neural additive deformation model provides the model with more structure. One could, of course, argue that more structure might in some cases limit the modeling capabilities. E.g., if nonlinear dependencies between covariates exist those would need to be explicitly added into a NAISR model. On the other hand, one could imagine that such nonlinear dependencies could be discovered by an unstructured neural network during the learning process.
>
>
> For very large sample sizes when shapes are densely sampled from the covariate space (as is the case for  many datasets used in general CV problems) such approaches can work well. However, when sample sizes are limited a more structured approach, as provided by $\texttt{NAISR}$, is likely more beneficial as the additive structure can help to better fit the model.  Specifically, in the neural additive design $\texttt{NAISR}$ uses individual networks to represent deformations driven by each covariate. The other covariates $\lbrace{c_{j, j \neq i}}\rbrace$ do not affect the individual network $g_i$ for $c_i$, and thus $g_i$ will explain   deformations based on only the covariate it is capturing. Take the pediatric airway as an example, the network for age will capture as best as it can  what shape effects can be attributed to age irrespective of  weight. As the models for the covariates are fit jointly they are of course indirectly linked as they jointly need to represent a given shape (for a particular age/weight combination), but each individual network is simpler than having one network that is targeted at capturing all possible dependencies. We believe this is the reason why our approach performs better than other methods for smaller sample sizes.   As in many realistic scientific shape analysis applications, such as our medical shape analysis problems, sample sizes are smaller our model should have high practical utility.
>
>
> We hope this helps answer your question. We are happy to provide further details and explanations if you like.

---

> > ### Comment · Reviewer_7YZF · 2023-11-18
> > **Thank you**
> >
> > Thanks!

---

### Official Review · Reviewer_juiL · 2023-10-30

**Soundness:** 3 good
**Presentation:** 4 excellent
**Contribution:** 3 good
**Rating:** 6
**Confidence:** 4

**Summary:**

The paper proposes a data model for representing shapes in a shape dataset as a function of given co-variates (i.e., age, weight, sex, etc.). It uses deep implicit representations to learn templates and deformations of shape specific to each co-variate whereas both the templates are deformations are not available apriori. The authors evaluated their method in terms of various shape analysis tasks (shape reconstruction, transfer, disentanglement, and evolution) by applying them to one simulated and two real human organ shape datasets. The experimental results support the claim of uniqueness and usefulness of the work.

**Strengths:**

+ The paper is a very good-read and tells a story in an elegant manner. It touches upon a new and potentially very impactful area of research in the domain of shape analysis of medical data.

+ The method is reasonable and the proposed claims are backed up by quantitative and qualitative evidence as experimental results

+ All the related works and their limitations have been comprehensively mentioned. The differences and similarities between the related works and the proposed work are clearly intelligible.

**Weaknesses:**

- The loss function is made up of 6 components, each associated with a scalar co-efficient to constitute the final loss. Though an ablation study on the loss components is presented in the supplementary table S.12 and S.13, they do not clearly discuss the effects of these co-efficients on the final result and what should be a reasonable set of values for them.

- Some of the limitations (invertibility problem) should be mentioned in the main manuscript instead of the supplementary. Ending the conclusion of the paper with a reference to the supplementary does not seem like a good approach to me.

- In Section 3.2, the authors simply mentioned that they used SIREN as the backbone of their template and deformation prediction network. I think a little bit more details on SIREN method needs to be added here.

**Questions:**

- All the shapes in the dataset seem to be pre-aligned and centered. Is it a requirement that the shapes need to be pre-aligned and centered? In such a case, it should be clearly mentioned in Problem Description (Section 3.1)

---

> ### Author Response · Authors · 2023-11-17
> **Answer to Reviewer 2**
>
> We thank the reviewer for the time, advice, and positive feedback. We address the main questions below:
>
> #### **The loss function is made up of 6 components, each associated with a scalar co-efficient to constitute the final loss. Though an ablation study on the loss components is presented in the supplementary table S.12 and S.13, they do not clearly discuss the effects of these co-efficients on the final result and what should be a reasonable set of values for them.**
>
> We thank the reviewer for this good question. We always set $\lambda_1=\lambda_5$ for $\mathcal{L}_{Eikonal}$. So there are 5 coefficients to be investigated. Doing a full grid search over all 5 parameters is prohibitive. Instead we can test random configurations from the full hyperparameter grid. We now tested 20 random configurations to 1) demonstrate that our setting can give good performance; and 2) provide guidance from failure settings. We now added this result to S.3.2 in the supplementary material. In summary, several good parameter settings exist, but the one we used is among the best.
>
> Specifically, we constructed a grid by multiplying our used settings ($\lambda_1 =\lambda_5 = 1 \cdot 10^1$; $\lambda_2 = 3 \cdot 10^1$; $\lambda_3 = 1 \cdot 10^1$, $\lambda_4 = 1 \cdot 10^2$, $\lambda_6 = \frac{2}{L}$; $\sigma=0.01$) for the 5 coefficients by $[0.01, 0.1, 1, 10, 100]$ to create a $5^5$ grid. Then, we randomly chose 20 grid points to test the model's robustness for different hyperparameter settings. Table S.15 and Table S.16 in the supplementary show the results of this analysis.
>
> The randomized grid-search analysis shows our hyperparameter setting is reasonable. We also performed an ablation study on top of this parameter setting (already contained in the original submission). Results of this ablation study are provided in Table S.12 and S.13.
> This ablation study shows that removing any of the reconstruction losses  ($\mathcal{L}\_{Eikonal}$, $\mathcal{L}\_{Dirichlet}$, $\mathcal{L}\_{Neumann}$) hurts performance.  A smaller $\lambda_6$ yields better reconstruction performance, but may hurt shape transfer performance.
>
>
> #### **Some of the limitations (invertibility problem) should be mentioned in the main manuscript instead of the supplementary. Ending the conclusion of the paper with a reference to the supplementary does not seem like a good approach to me**
>
> Thanks for the suggestions. We will try our best to create some space and update our manuscript accordingly.
>
> #### **In Section 3.2, the authors simply mentioned that they used SIREN as the backbone of their template and deformation prediction network. I think a little bit more details on SIREN method needs to be added here.**
>
> Thanks for the suggestions. We have updated the following details to S.3.1 in our manuscript.
>
> We follow SIREN for the network setting and parameter initialization. SIREN proposes to use periodic activation functions for implicit neural representations and demonstrates that networks, which use periodic functions (such as sinusoidal functions) as activations, are ideally suited for representing complex natural signals and their derivatives. We also follow SIREN's initialization to draw weights according to a uniform distribution $\mathcal{W} \sim \mathcal{U}\left(-\sqrt{6 / D_{in}}, \sqrt{6 / D_{out}}\right)$, ($D_{in}$ and $D_{out}$ are input and output dimensions).
>
>
> #### **All the shapes in the dataset seem to be pre-aligned and centered. Is it a requirement that the shapes need to be pre-aligned and centered? In such a case, it should be clearly mentioned in Problem Description**
>
> This is a good question. We mentioned in Supplementary S.2 that the hippocampus shapes are rigidly aligned using the ICP algorithm. The airway shapes are rigidly aligned using the anatomical landmarks. We now mention this in the Problem Description (Section 3.1). Thanks for the suggestion.

---

> > ### Comment · Reviewer_juiL · 2023-11-22
> >
> > I would like to thank the authors for a reasonable response that addressed most of my concerns. I would suggest the authors incorporate a summary of the responses (mainly limitations) in their final manuscript.

---

> > > ### Author Response · Authors · 2023-11-23
> > > **Thank you**
> > >
> > > Thanks for your timely response and excellent suggestion! We are happy to incorporate such a summary in our final paper.

---

> ### Author Response · Authors · 2023-11-22
> **Kind Reminder**
>
> Dear Reviewer juiL,
>
> $~$
>
> Thank you again for your valuable feedback and comments! As the discussion period is approaching its end in 18 hours, we would greatly appreciate it if you could let us know whether our rebuttal and the updated manuscript addressed your concerns and if so, if it's possible to increase your rating. We are happy to address any of your remaining concerns.
>
>
> $~$
>
> Sincerely,
>
> Authors of Paper 2838

---

### Official Review · Reviewer_ScLH · 2023-11-01

**Soundness:** 3 good
**Presentation:** 3 good
**Contribution:** 3 good
**Rating:** 6
**Confidence:** 3

**Summary:**

This paper proposes a 3D Neural Additive Model for Interpretable Shape Representation (NAISR) which describes individual shapes by deforming a shape atlas in accordance to the effect of disentangled covariates. The proposed approach captures shape population trends and allows for patient-specific predictions through shape transfer. Moreover, NAISR is the first approach to combine the benefits of deep implicit shape representations with an atlas deforming according to specified covariates. Sufficient experiments demonstrate that NAISR achieves excellent shape reconstruction performance while retaining interpretability.

**Strengths:**

1. The idea is ok
2. This paper is well-written and easy to follow
3. The experiment is sufficient

**Weaknesses:**

1. The research survey on the investigated model is relatively limited.
2. The construction details of the algorithm need to be improved.

**Questions:**

1. It would be better if the authors give more details of how to train the additive model $g_i$?
2. Does the inclusion of group structure contribute to the improvement of model performance? Please refer to the Group Sparse Additive Model (Yin et al 2012; Chen et al 2017).

---

> ### Author Response · Authors · 2023-11-17
> **Answer to Reviewer 1**
>
> We thank the reviewer for the time, advice, and positive feedback. We address the main questions below:
>
>
> #### **The research survey on the investigated model is relatively limited.**
>
> Due to the 9-page limit, we were only able to include part of the research survey in the main text. We selected what we thought might be the most relevant material. A more comprehensive research survey is available in the supplementary material S.1., including discussions of **deep implicit functions**, **point correspondences**, **disentangled representation learning**, **articulated shapes**, and **explainable artificial intelligence**, as well as how these methods relate to yet differ from our method.
>
> We will perform an additional literature review on the most recent related work and update our manuscript accordingly. We are happy to accommodate any suggestions related to moving survey material from the supplementary material to the main document (subject to space constraints) or the inclusion of missing related works.
>
>
> #### **The construction details of the algorithm need to be improved**
>
> We are sorry about not placing enough construction details in the main text. In the supplementary material (S.3.1), we include the implementation details, such as parameter settings and data preprocessing steps. We now added more details about model construction. We now also added Figure.S.5 to the supplementary material for a clearer illustration of our approach. We will also make our implementation publicly available on *GitHub* so others can reproduce and build upon our results. See also our responses to the two specific construction questions below.
>
> #### **It would be better if the authors give more details on how to train the additive model ${g}_i$**
>
>
> Each subnetwork, including the template network $\mathcal{T}$ and the displacement networks $\{f_i\}$, are all parameterized with an $N_l$-layer MLP using $sine$ activations. We use $N_l=8$ for Starman and the ADNI hippocampus dataset; we use $N_l=6$ for the pediatric airway dataset. There are 256 hidden units in each layer. The architecture of the $\{f_i\}$ follows DeepSDF, in which a skip connection is used to concatenate the input of $(\mathbf{p}, c_i)$ to the input of the middle layer, as shown in the added Figure.S.5 in the supplementary material. We use a latent code $\mathbf{z}$ of dimension 256 ($L=256$). We follow SIREN for the network setting and parameter initialization.
>
> For each training iteration, the number of points sampled from each shape is 750 ($N=750$), of which 500 are on-surface points ($N_{on} = 500$) and the others are off-surface points ($N_{off} = 250$). We train $\texttt{NAISR}$ for 3000 epochs for the airway dataset and for 300 epochs for the ADNI hippocampus and Starman datasets using Adam with a learning rate $5e-5$ and batch size of 64.
>
> Also, we jointly optimize the latent code $\mathbf{z}$ with $\texttt{NAISR}$ using Adam with a learning rate of $1e-3$. We use a latent code $\mathbf{z}$ of dimension 256 ($L=256$). During training, $\lambda_1 =\lambda_5 = 1 \cdot 10^1$; $\lambda_2 = 3 \cdot 10^1$; $\lambda_3 = 1 \cdot 10^1$, $\lambda_4 = 1 \cdot 10^2$. For $\mathcal{L}_{lat}$, $\lambda_6 = \frac{2}{L}$; $\sigma=0.01$ (following DeepSDF). During inference, the latent codes are optimized for $N_t$ iterations with a learning rate of $5e-3$. $N_t$ is set to 800 for the pediatric airway dataset; $N_t$ is set to 200 for the Starman and ADNI Hippocampus datasets.
>
> These details are included in S.3.1 in the supplementary material.
>
> #### **Does the inclusion of group structure contribute to the improvement of model performance? Please refer to the Group Sparse Additive Model (Yin et al 2012; Chen et al 2017).**
>
> Thanks for the excellent suggestion of discussing group structure in covariates. We will include the suggested references and add a brief forward-looking discussion for future work. Note that our experiments are currently only based on very few covariates: two for the Starman dataset, three for the airway dataset, and four for the hippocampus dataset. In contrast, Yin et al.'s work targets 8,141 genes, and Chen et al. uses on the order of 20 covariates in their synthetic and real experiments. Exploring how group sparsity could be included in our NAISR model for high-dimensional covariates (for example, to explore the interaction between shape and gene expression on datasets that provide both datatypes) and how it would improve results would be extremely interesting future work.

---

> ### Author Response · Authors · 2023-11-22
> **Kind Reminder**
>
> Dear Reviewer ScLH,
>
> $~$
>
> Thank you again for your valuable feedback and comments!
> As the discussion period is approaching its end in 18 hours, we would greatly appreciate it if you could let us know whether our rebuttal and the updated manuscript addressed your concerns and if so, if it's possible to increase your rating. We are happy to address any of your remaining concerns.
>
> $~$
>
> Sincerely,
>
> Authors of Paper 2838

---

### Author Response · Authors · 2023-11-22
**Notice of Our Revision**

Dear reviewers,

$~$

We appreciate your timely feedback and excellent suggestions!

Based on your suggestions and our discussions, we revised our manuscript with the revisions marked blue. Specifically,

* We performed additional literature reviews, mainly on neural deformable models for point correspondences. We selected the most relevant material in the main text and made a more extended version in the S.1 in the supplementary material.
* We added discussions about limitations and future works about invertibility, the inclusion of abnormal cases, and group sparsity in the main text.
* We added more implementation details in S.3.1 in the supplementary material, mainly including the setting of SIREN, computational runtime, and network design.
* We added more ablation studies of hyperparameter settings in S.3.2 in the supplementary material.
* We visualized the template learning process during the training stage in S.3.6 in the supplementary material.

We hope you enjoy it!
Many thanks in advance!

$~$



Best,

Authors of Paper 2838

---

### Meta-Review · Area_Chair_e9Ac · 2023-12-09

**Metareview:**

The paper introduces a method for representing and predict the shapes of anatomical objects using implicit neural representations which deform an atlas based on covariates, enabling interpretability. The reviewers unanimously recommend acceptance.

**Justification For Why Not Higher Score:**

Limited real world evaluation beyond anatomical datasets.

**Justification For Why Not Lower Score:**

Well written, well motivated method with good evaluation of performance.

---

### Decision · Program_Chairs · 2024-01-16

Accept (spotlight)